

# Hydrological Concept Formation inside Long Short-Term Memory (LSTM) networks

Thomas Lees[1,5], Steven Reece[2], Frederik Kratzert[6], Daniel Klotz[3], Martin Gauch[3], Jens De Bruijn[5,7], Reetik Kumar Sahu[5], Peter Greve[5], Louise Slater[1], and Simon J. Dadson[1,4]

[1]School of Geography and the Environment, University of Oxford, South Parks Road, Oxford, United Kingdom, OX1 3QY
[2]Department of Engineering, University of Oxford, Oxford, United Kingdom
[3]LIT AI Lab & Institute for Machine Learning, Johannes Kepler University Linz, Linz, Austria,
[4]UK Centre for Ecology and Hydrology, Maclean Building, Crowmarsh Gifford, Wallingford, United Kingdom, OX10 8BB
[5]International Institute for Applied Systems Analysis (IIASA), Laxenburg, Austria
[6]Google Research, Vienna, Austria
[7]Institute for Environmental Studies, VU University, De Boelelaan 1087, 1081HV, Amsterdam, The Netherlands

**Correspondence:** Thomas Lees (thomas.lees@chch.ox.ac.uk)

**Abstract.** Neural networks have been shown to be extremely effective rainfall-runoff models, where the river discharge is predicted from meteorological inputs. However, the question remains, what have these models learned? Is it possible to extract information about the learned relationships that map inputs to outputs? And do these mappings represent known hydrological concepts? Small-scale experiments have demonstrated that the internal states of Long Short-Term Memory Networks (LSTMs),

a particular neural network architecture predisposed to hydrological modelling, can be interpreted. By extracting the tensors which represent the learned translation from inputs (precipitation, temperature) to outputs (discharge), this research seeks to understand what information the LSTM captures about the hydrological system. We assess the hypothesis that the LSTM replicates real-world processes and that we can extract information about these processes from the internal states of the LSTM. We examine the cell-state vector, which represents the memory of the LSTM, and explore the ways in which the LSTM learns

to reproduce stores of water, such as soil moisture and snow cover. We use a simple regression approach to map the LSTM state-vector to our target stores (soil moisture and snow). Good correlations (R2 > 0.8) between the probe outputs and the target variables of interest provide evidence that the LSTM contains information that reflects known hydrological processes comparable with the concept of variable-capacity soil moisture stores.

The implications of this study are threefold: 1) LSTMs reproduce known hydrological processes. 2) While conceptual

models have theoretical assumptions embedded in the model a priori, the LSTM derives these from the data. These learned representations are interpretable by scientists. 3) LSTMs can be used to gain an estimate of intermediate stores of water such as soil moisture. While machine learning interpretability is still a nascent field, and our approach reflects a simple technique for exploring what the model has learned, the results are robust to different initial conditions and to a variety of benchmarking experiments. We therefore argue that deep learning approaches can be used to advance our scientific goals as well as our

predictive goals.





## 1 Introduction

LSTMs have demonstrated state-of-the-art performance for rainfall-runoff modelling for a variety of locations and tasks (Kratzert et al., 2018, 2019c; Ma et al., 2020; Lees et al., 2021; Frame et al., 2021). However, whether we can use these models to better interpret the hydrological system remains an open question. Given that LSTM-based models offer state-of-the-art

hydrological performance, more research is required to better understand what conceptual structures the LSTM has learned and to diagnose potential gaps in our conceptual and process-based models, ultimately to stimulate innovation in hydrological theory.

The primary objective of this study is to test the hypothesis that the information stored in the LSTM state-vector reflects known hydrological concepts that are important for discharge generation, including soil water storage and snow processes.

What have these models learned about the hydrological system that allows them to make highly accurate predictions? Can we interrogate the model to determine whether the LSTM has learned a physically realistic mapping from inputs to outputs? Being able to reason about the model and its behavior is a key component of dependable models. It allows researchers and practitioners to interrogate the model, making sure that it is giving the right results for the right reasons (Kirchner, 2006).

Deriving insights about the hydrological system has always been a goal of hydrological modelling (Beven, 2011). Peter

Young's work on Data-Based Mechanistic modelling (DBM) emphasised the need to apply flexible data-driven models before then applying a mechanistic interpretation to the learned representation of these models (Young and Beven, 1994; Young, 2003, 1998). Philosophically, this approach is similar to the one we take here, although the number of parameters in the DBM approach is much smaller. In an early application of neural networks to rainfall-runoff modelling, Wilby et al. (2003) sought to challenge preconceptions of neural network approaches as uninterpretable. They found that nodes in their Multi-

Layer Perceptron corresponded to quickflow, baseflow and soil saturation, and showed how the learned representation of deep learning models could be interpreted. They sought to determine whether neural networks were capable of reproducing both the outputs and internal functioning of conceptual hydrological models.

Recent studies call to more fully explore the potential for techniques from the fields of artificial intelligence and machine learning (Beven, 2020; Reichstein et al., 2019; Shen, 2018; Karpatne et al., 2017) by demonstrating predictive performance

alongside interpretations of the model itself to improve our understanding of the modelled system. Several studies have suggested that LSTM rainfall-runoff models learn a generalizable representation of the underlying physical processes. This allows them to perform well in out-of-sample conditions, such as Prediction in Ungauged Basins (PUB) (Kratzert et al., 2019b; Feng et al., 2020; Ma et al., 2020), and unseen extreme events (Frame et al., 2021). These results suggest that LSTMs have captured information that generalizes to these conditions, information that can help us improve hydrological theory and predictions.

Outside of hydrology, calls for interpreting machine learning and deep learning systems are getting louder and even generating legislative changes (European Union Digital Strategy, 2019; UK Statistics Authority, 2019). Spiegelhalter (2020), for example, argues that as algorithmic decision support tools become widespread in everyday life, the ability to describe how predictions are made is essential for building trust in these systems. A large body of literature has arisen to: define interpretability (Ribeiro et al., 2016; Lipton, 2018; Doshi-Velez and Kim, 2017); to measure how interpretable models aid human decision



making (Nguyen, 2018; Chu et al., 2020); and to develop methods for interpreting models (Olah et al., 2018, 2020; Ghorbani and Zou, 2020; Lundberg and Lee, 2017). Our contribution draws on work from neuro-linguistic programming, where learned embeddings from models trained for speech-recognition tasks have been interpreted to better understand how parts-of-speech are recognised and used by LSTM models (Hewitt and Liang, 2019).

The exploration of the internal representations of LSTM based rainfall-runoff models is still at an early developmental stage. Kratzert et al. (2018) showed evidence that individual LSTM cells correlate with snow water content, although the model was only trained to predict discharge from meteorological inputs. Kratzert et al. (2019c) explored the learned embedding of catchment attributes, showing that an LSTM variant had learned to group the rainfall-runoff behaviours of hydrologically similar catchments. Using dimensionality reduction techniques, the static embedding (the output of the input gate) was shown to reflect spatial and thematic groups of catchments that qualitatively correspond to catchments with similar hydrological behaviours. For two exemplary basins, Kratzert et al. (2019a) found correlations between the cell states of the LSTM and three hydrological states (upper zone storage, lower zone storage and snow depth) from the Sacramento + Snow-17 hydrological model (Burnash et al., 1995). All three of these studies introduced methods that can be used for exploring the internal representation of hydrological models, but there exists no comprehensive evaluation of the information stored in the cell state dimensions across a large sample of basins. Furthermore, these studies only compared individual memory cells with hydrological processes. The LSTM however, is not forced to store information about one process in a single memory cell but can distribute the information about hydrological processes across several cells. Therefore, we explore methods for extracting the information that is stored in the LSTM cell state, across all cells.

The aim of this research is to examine the internal functioning of the LSTM model. We explore the evolution of the LSTM state-vector and test whether information that reflects intermediate stores of water (soil moisture and snow depth) has been learned by the LSTM. This research is novel for providing a means of interpreting what information the LSTM rainfall-runoff model has encoded within its state-vector. To our knowledge, we are the first to apply techniques developed in machine learning interpretability and natural language processing research (Hewitt and Liang, 2019) to hydrology. We carry out a comprehensive evaluation of the LSTM cell states across a sample of 669 catchments in Great Britain (Lees et al., 2021). This allows us to rigorously assess whether the LSTM has learned concepts that generalize over space. On this basis we devised several baseline experiments to provide evidence for an internal representation of hydrologically relevant processes. Furthermore, we consider information stored across all values in the LSTM state-vector, as opposed to identifying and focusing only on single values from within the cell state. This is important since there are no constraints forcing the LSTM to store information in individual cells.



## 2 Methods

In this study, we trained LSTM models using the same hyper-parameters as those trained in Lees et al. (2021). We offer a brief introduction to the state-space formulation of the LSTM (Kratzert et al., 2019a) because it offers a clear explanation for why we explore the cell-state ($c_t$), since it reflects the state-vector of the LSTM.

### 2.0.1 The LSTM

Hydrological models are often formulated with a state-space based approach. This means that the states ($s$) at a specific time ($t$) depend on the input at time t ($x_t$), the model state in the previous timestep ($s_{t-1}$) and the model parameters ($\theta$) (Kratzert et al., 2019a).

$$s_t = g(i_t, s_{t-1}; \theta_j) \tag{1}$$

The model output ($y_t$, discharge) is a function of the states ($s_t$) and inputs ($i_t$) at that timestep, and the model parameters.

$$y_t = g(i_t, s_t; \theta_j) \tag{2}$$

Similarly, the LSTM can be formulated as:

$$c_t, h_t = f_{\text{LSTM}}(x_t, c_{t-1}, h_{t-1}; \theta_k) \tag{3}$$

$$y_t = f_{\text{Dense}}(h_t; \theta_l) \tag{4}$$

Where the state-vector (the "cell state" $c_t$) and output-vector (the "hidden state" $h_t$) of the LSTM at timestep t are a function of the current inputs ($x_t$, e.g. meteorological features and catchment attributes), the previous output and state ($h_{t-1}$ and $c_{t-1}$) and some learnable parameters ($\theta_k$). Similar to the state-update equations, the output of the model ($y_t$, e.g. the discharge) is a function of the output of the LSTM ($h_t$, which is a function of $c_t$) and some more (learnable) model parameters ($\theta_i$).

The key difference between the LSTM and classical state models (e.g. conceptual and physical hydrology models) is that the LSTM can infer any process that is deducible from the data to solve the training task, while classical hydrological models are limited by the processes that are hard-coded in the model implementation.

In order for the LSTM models to produce accurate simulations of discharge across a variety of catchments, we hypothesise that the LSTM should have learned to represent hydrological processes and stores. We test whether the LSTM is able to recover intermediate stores of water by visualising the evolution of the LSTM cell-state and compare this to soil moisture and snow depth from ERA5-Land.

### 2.1 Experimental Design

We used the following experimental design to investigate the learned hydrological process understanding of LSTMs.

Following Lees et al. (2021), we trained a single LSTM to predict runoff for 669 basins from the CAMELS-GB dataset (Coxon et al., 2020b). The input sequences are digested into the LSTM each consisting of one year's worth of daily data (365





timesteps). The model is forced by a set of meteorological variables (precipitation and temperature) and a series of static catch-
ment attributes describing topography, climatic conditions, soil types and land cover classes. These static attributes are used
to learn differences and similarities between catchments. For more details of the training procedure and for a comprehensive
table listing all model inputs, we refer the reader to Table 2 of Lees et al. (2021). It is important to note that neither snow depth
nor soil moisture were included as inputs or outputs during model training.

## 2.2  Probing

In the present context, a probe is a diagnostic device that is used on top of the trained LSTM model to examine the learned
internal representation of the LSTM. In its simplest form, a probe is a linear regression model that connects the cell states to a
given output. In a more complex form a probe might be realized in the form of a set of stacked multi-layer perceptrons, or any
other algorithm fit for regression tasks. As such, probes offer the opportunity to explore what the LSTM has learned during
training, allowing us to use the LSTM to generate predictions of latent, intermediate variables. They also confirm whether
our model has learned physically realistic mappings from inputs to outputs. To our knowledge, probes have not been used on
hydrological LSTMs.

Probes have been used in natural language processing tasks to determine whether learned embeddings in deep learning mod-
els contain information that can be matched to semantically meaningful concepts (Hewitt and Liang, 2019). The embeddings
are used as inputs, and a probe is trained to map these embeddings onto properties, such as part-of-speech tags.

Since these probes are trained in a supervised way, we are currently limited to looking for known hydrological processes.
Trained in this way, probes cannot be used to extract unknown information, since we require a target variable to fit the probe.
This means that in the present study we are not looking for new hydrological understanding, or seeking to uncover as-of-yet
undiscovered hydrological patterns, but explicitly looking for known physical processes in the learned LSTM representation.
This paper demonstrates a conceptual innovation moving the field towards extracting information from the LSTM, diagnosing
whether these neural networks are learning physically realistic processes.

In this paper, we use the probe to explore whether the LSTM state-vector has information that is predictive of different latent
hydrological variables, such as soil moisture at various depths and snow water equivalent. We begin with the simplest probe,
a linear model. This encodes the strong assumption that latent variables can be extracted as a linear combination of the cell
state values. While this is not necessarily the case, for the purposes of probe interpretation the simplicity of the linear model
is preferred. It allows for intuitive explanations and simple visual analysis of results by exploring the weights of the linear
model, and by interpreting the probe predictions. For our experiments, we fit one probe for all catchments, learning a set of
weights and a bias term for each target variable (Figure 1c, d). That is, we hypothesize that there is a common set of weights
that generalize to all basins in the training set.

As a control experiment, we ensure that we are not learning spurious relationships between the LSTM state and intermediate
hydrological stores (soil moisture and snow depth) by designing two experiments to test that our findings are specific in time
and space. We test both shuffling the target variables in space (i.e. breaking the spatial link between cell-states and the target
variable) and shifting the target variable in time (breaking the temporal link between inputs and outputs), and find that these



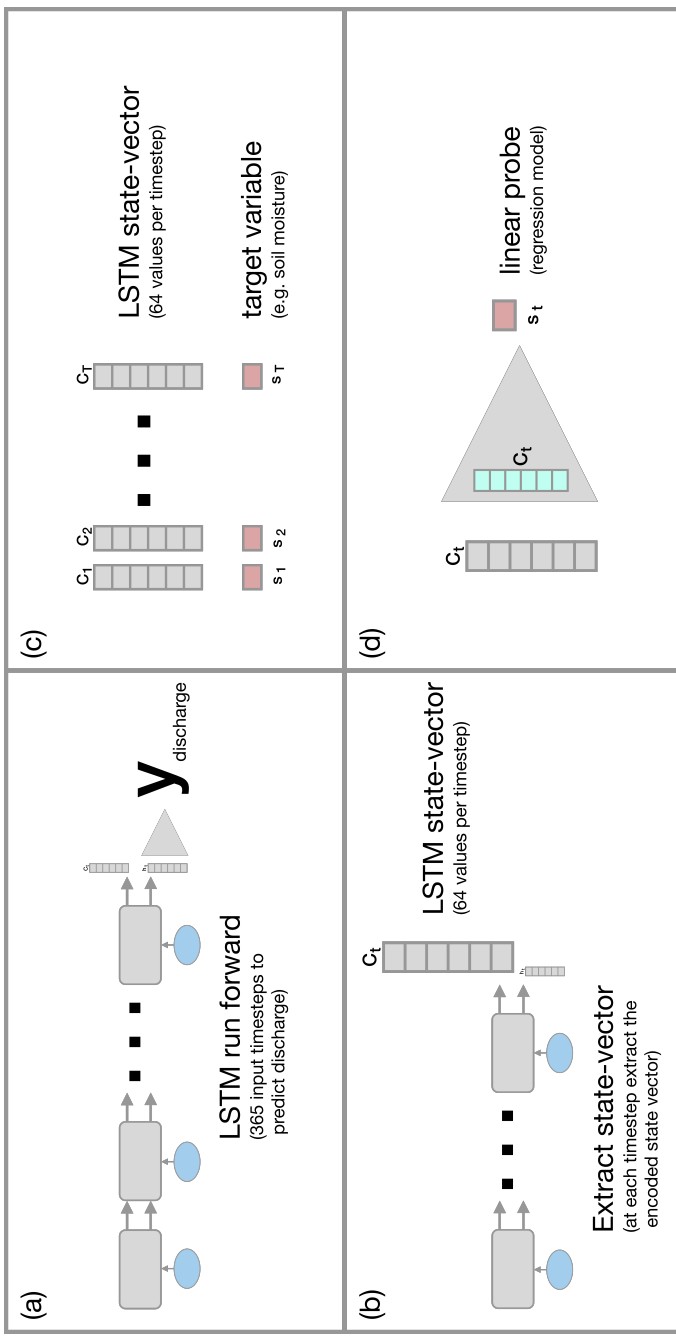

**Figure 1.** An overview of the linear probe analysis. (a) Demonstrates the LSTM as an input-state-output model, where at each timestep, inputs (blue spheres, such as precipitation) are processed producing a prediction of discharge on the 365th day. This reflects how the LSTM is trained. (b) When forcing the model with already trained weights, we extract the LSTM state vector ($c_t$) at each timestep. (c) We then compile a dataset of inputs (the $c_t$ vectors for each target timestep) and targets ($s_t$ - the soil moisture measurements for the catchment) matched at each catchment and timestep. (d) Finally, we use this dataset to train a linear probe, a set of weights and a bias term for all catchments.





results show that the information contained in the cell-state vector is specific to the given location and time. These results can be found in Appendix Sect. A.

We train a linear probe parameterised by $\beta$ to make predictions of our target storage variable ($\hat{s}$) for each catchment ($i$) and at each timestep ($\{1:T\}$).

$$\hat{s}_{i,\{1:T\}} = f_\beta(c_{i,\{1:T\}}) \tag{5}$$

Our linear model, $f_\beta$, is a penalized linear regression model. We use the elastic-net regularisation that combines the $\ell_1$ penalty of lasso regression with the $\ell_2$ penalty of ridge regression. The reason for choosing the elastic-net regularisation is that

when we have correlated features in $c_t$, the lasso regression is likely to pick one of these at random, whereas the elastic-net regression will assign weights to both (Friedman et al., 2010). The lasso ($\ell_1$ penalty) shrinks non-informative weights to zero, and the ridge ($\ell_2$ penalty) ensures that correlated features are not randomly set to zero by the lasso ensures that the model is stable under rotation, so both are useful. The objective function becomes:

$$\min_\beta \frac{1}{2n_{\text{samples}}}||c_{i,t}\beta - y||_2^2 + \alpha\rho||\beta||_1 + \frac{\alpha(1-\rho)}{2}||\beta||_2^2 \tag{6}$$

We set the $\alpha$ parameter to 1.0 (describing the degree of shrinkage) and the $\rho$ parameter to 0.15 (describing the degree of $\ell_1$ loss relative to $\ell_2$ loss). These parameters were set in order to give the best training-sample performance and ensure that non-informative weights are shrunk to zero.

## 2.3 ERA5-Land Data

In order to determine whether the information stored in the LSTM state-vector reflects known hydrological concepts, we used

variables from the ERA5-Land dataset as the probe targets (Figure 1c, d). As designed, this protocol will determine whether the LSTM has learned consistent representations by comparing the probe output to commonly used soil-moisture products, including reanalysis data, rather than concepts directly from in-situ observations. Reanalysis observations were preferred because of the longer time series of available data, the gridded form meaning that a catchment-averaged soil moisture can be calculated, and because these products are globally available, meaning that this approach would generalise to LSTMs trained

elsewhere. However, there are uncertainties associated with the soil moisture estimates from these gridded reanalysis products.

We used soil moisture and snow-depth data from ERA5-Land (Muñoz-Sabater et al., 2021), a reprocessing of the land surface components of ERA5 forced by the ERA5 atmospheric model (Hersbach et al., 2020). ERA5-Land is a reanalysis product with 9 km grid spacing and an hourly temporal frequency. It is a global land surface reanalysis dataset for describing the water and energy cycles. Muñoz-Sabater et al. (2021) demonstrate that the ERA5-Land product has improved soil moisture

and snow observations compared with ERA-Interim products when evaluated against in-situ soil moisture measurements. The results when compared against ERA5 are more variable, although ERA5-Land does show improvements in North America and small improvements in Europe. ERA5-Land does not assimilate soil moisture observations directly, rather the assimilation of observed meteorological data occurs only in the calculation of the atmospheric forcing variables from ERA5. Therefore, we can be confident that ERA5-Land provides a calculation of soil moisture independent of the observational, catchment-averaged





datasets included in CAMELS-GB (Coxon et al., 2020b) which are used to train the LSTM. It is worth mentioning that the main focus of this study is to test whether the LSTM models intermediate stores of water (i.e. matches the relative changes in that unseen variable over time), and not necessarily to compare the values to the best-possible soil moisture (or snow) estimate/measurement.

The soil moisture data in ERA5-Land contains four layers, each of which is used as a target variable. The top layer, soil water
volume level 1 (swvl1), is from 0-7cm, the second layer (swvl2) from 7-28cm, the third layer (swvl3) from 28-100cm and the final layer (swvl4) from 100-289cm. We therefore fit four separate probes, using each soil moisture layer as an independent target variable.

One drawback of using reanalysis soil moisture (ERA5-Land) is that we identify modelled soil moisture, rather than a directly observed soil moisture signal - that is, we discover whether our LSTM model has learned a process representation. To
counter this, we also assess probe performances on alternative products such as ESA CCI Soil Moisture, a blended product combining multiple satellite-derived soil moisture estimates (see Appendix Sect. B). The ESA CCI dataset comes with its own caveats, namely that the measured soil moisture is restricted to the top-layer (5-10cm) and so we cannot observe whether deeper layers are represented by the internal state of the LSTM. Furthermore, the depth of the satellite derived estimate is itself a function of the water content and surface roughness. ESA CCI is not a direct observation either, but a blended estimate using
radiative transfer functions and microwave-based estimations of soil moisture. We use ERA5-Land soil moisture and snow depth as our target variables for the results section of this paper. Further comparisons against ESA CCI Soil Moisture can be found in Appendix B.

We clipped the probe target variables (ERA5-Land soil water layers 1, . . . , 4 and snow depth) to the catchment shapefiles provided as part of CAMELS-GB dataset (Coxon et al., 2020a) and calculated a catchment mean to produce a lumped catch-
ment soil moisture time-series. This follows the methodology used to generate the CAMELS-GB meteorological forcing data. In order to train the linear probe we normalize the target data (ERA5-Land), centering the data using the mean target value across every catchment, and rescaling the data using the standard deviation of the target data. Both statistics are calculated in the training period.

## 3   Results

### 3.1   Soil Moisture Probe

We use the linear probe described above to see if a learned soil moisture signal for four soil-depths (from ERA5-Land) is present in the LSTM. The correlation between the inferred soil water volume and the target data from ERA5-Land is shown by the histograms in Fig. 2. We can see that on average the inferred soil moisture for the upper three soil layers has a high correlation with the soil moisture from ERA5-Land (median scores of 0.85, 0.90 and 0.89 for level 1, level 2 and level 3
respectively). The median catchment correlation coefficient for the fourth soil layer is less than the three upper layers (0.77). However, we would argue that for all four soil layers, the LSTM state seems to model the dynamics of the soil water content. For reference, we have run two experiments that act as a baseline (Appendix A).





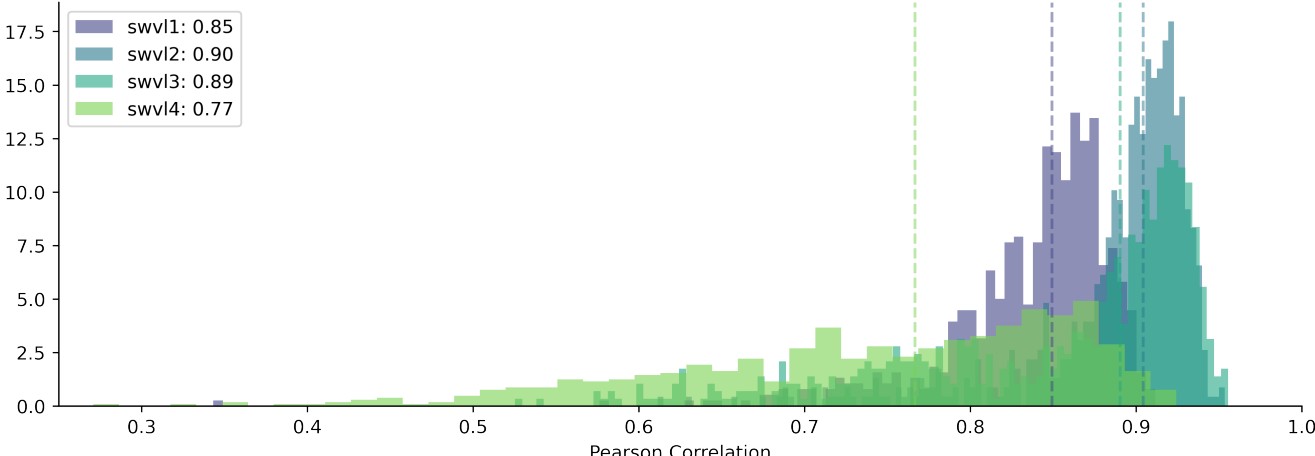

**Figure 2.** Histogram of catchment correlation scores for each soil water volume level: soil water volume level 1 (swvl1) contains soil moisture from depths of 0–7cm, soil water volume level 2 (swvl2) contains soil moisture from depths of 7–28cm, soil water volume level 3 (swvl3) contains soil moisture from depths of 28–100cm, soil water volume level 4 (swvl4) contains soil moisture from depths of 100–289cm.

The time series plots in Fig. 3 show that the linear probe is able to capture the dynamics of the soil moisture values. However, modelling the catchment-specific offset in catchments with more or less saturated soils than the GB mean is difficult with the linear probe (e.g. Fig. 3b, a catchment with less saturated soils than the GB mean has an observed soil water volume less than the zero point, which describes the GB mean). By offset, we are referring to the point around which soil moisture fluctuates, the mean saturation level for that catchment. Looking at Fig. 4 confirms this hypothesis, since the catchments with the largest biases in probe outputs (blue circles) have wetter and drier than average conditions, compared with the smallest biases (orange), which are concentrated in the middle of the distribution.

As we mentioned in Sect. 2.3, we centered and rescaled the ERA5-Land soil moisture data using the global mean and standard deviation. So a value of zero corresponds to the mean soil water volume across all catchments in the training period. This can be seen in catchment 15021, where the catchment specific soil moisture level is below zero (the grey dashed line is below zero), but the probe continues to predict values centered on zero. The dynamics remain well modelled (and therefore correlation scores are high), but the probe identifies soil moisture anomalies rather than absolute values. Ultimately, we should expect this behaviour since we are fitting a single linear model with only one bias term. For alternative methods that model catchment-specific offsets, please see the experiment including a catchment-specific bias term (by including the gauge_id as input to the model, Appendix Sect C) and the experiments with a non-linear model (Appendix Sect. E). We return to this point in the discussion that follows.

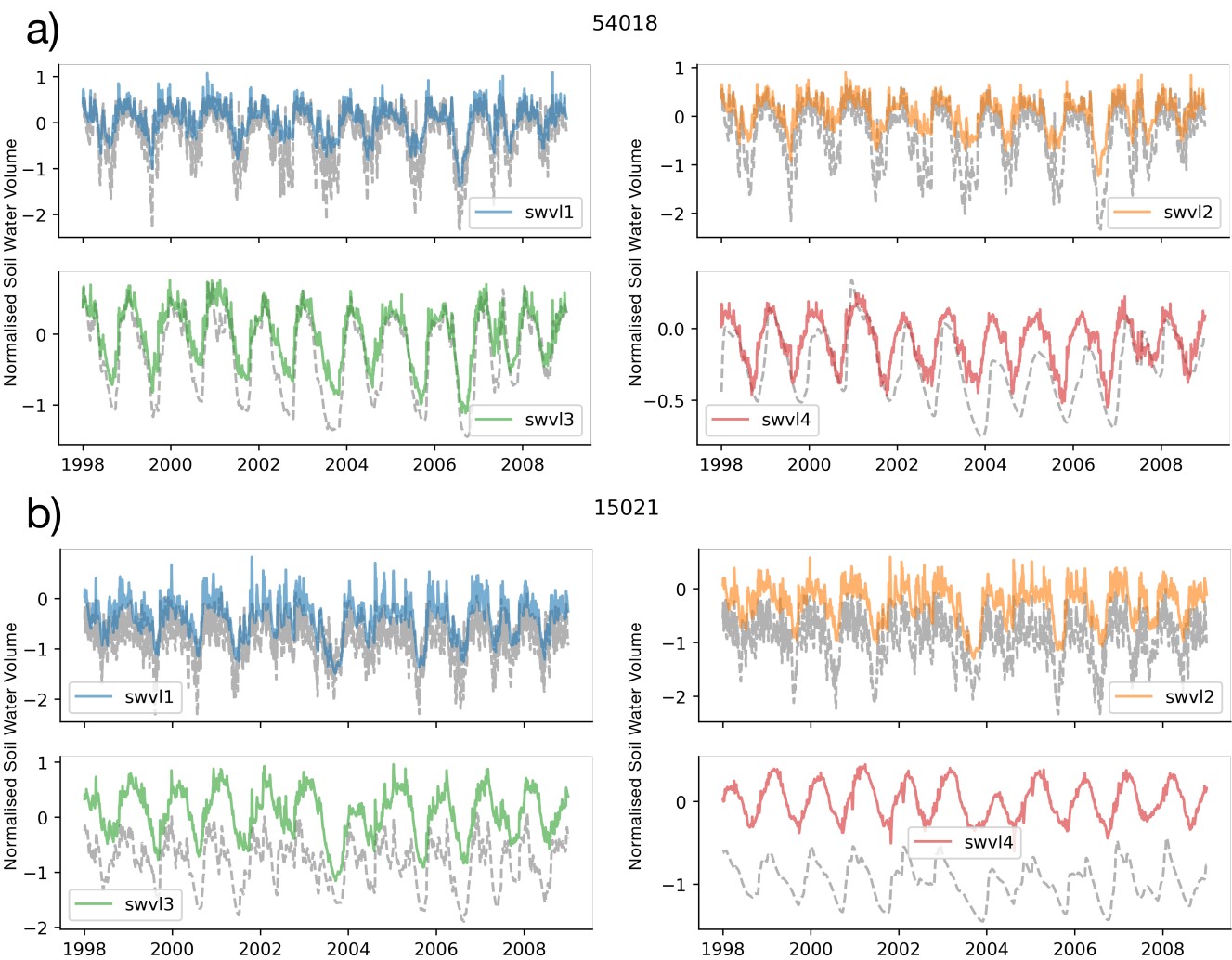

**Figure 3.** Time series of probe predictions (coloured lines) compared with the target variables (grey dotted lines). We show two catchments here, (a) 54018 - Rea Brook at Hookagate and (b) 15021 - Burn at Burnham Overy (chosen to reflect both a catchment with a small bias and a catchment with a strong bias, see Appendix D for more information on these two catchments) and four soil moisture levels, swvl1 (blue), swvl2 (orange), swvl3 (green), swvl4 (red). The probe captures the temporal dynamics of the soil moisture signals, but shows systematic bias, consistently predicting variability about zero, which defines the mean GB-wide soil moisture.





**Figure 4.** a) The spatial location of the 60 catchments with the largest probe biases (blue) and smallest probe biases (orange) (b) The distribution of catchment log mean precipitation for the catchments with the largest and smallest probe biases. The largest 60 biases (blue) occur in catchments that are wetter or drier than average, as demonstrated by the gap in the middle of the distribution of mean catchment precipitations. By contrast the smallest 60 biases (orange) are mostly concentrated in the middle of the distribution, as we expect when using a simple linear model as the probe. These distributions are significantly different when using a 2-sample Kolmogorov-Smirnov test.





## 3.2 Snow Depth Probe

Another process that influences river discharge is snow water storage. ERA5-Land offers a snow depth variable (m of water equivalent) that serves as a proxy of snow water equivalent. In order to determine whether the LSTM is representing snow processes in the state-vector, we use the probe analysis with ERA5-Land snow depth as our target variable.

Since snow processes are only significant in very few basins in Great Britain, we trained the probe on a subset of the basins shown in Figure 5c. These are defined as those catchments with a proportion of precipitation falling as snow greater than 5%.

Figure 5b shows the probe output over one year for one station (Station 15025, Ericht at Craighall) and then Fig. 5a shows the probe output for that station and two other snowy catchments over the entire test period (1998 – 2008). They show clearly that the probe output correctly predicts very little variability in the summer period, when the snow will have melted. The winter peaks correspond to snow accumulation and snow melt. The median (over 33 catchments) Pearson correlation coefficient between the probe simulated snow depth and the ERA5-Land snow depth is 0.84, meaning that 84% of the snow variance can

be described by the linear probe simulation.

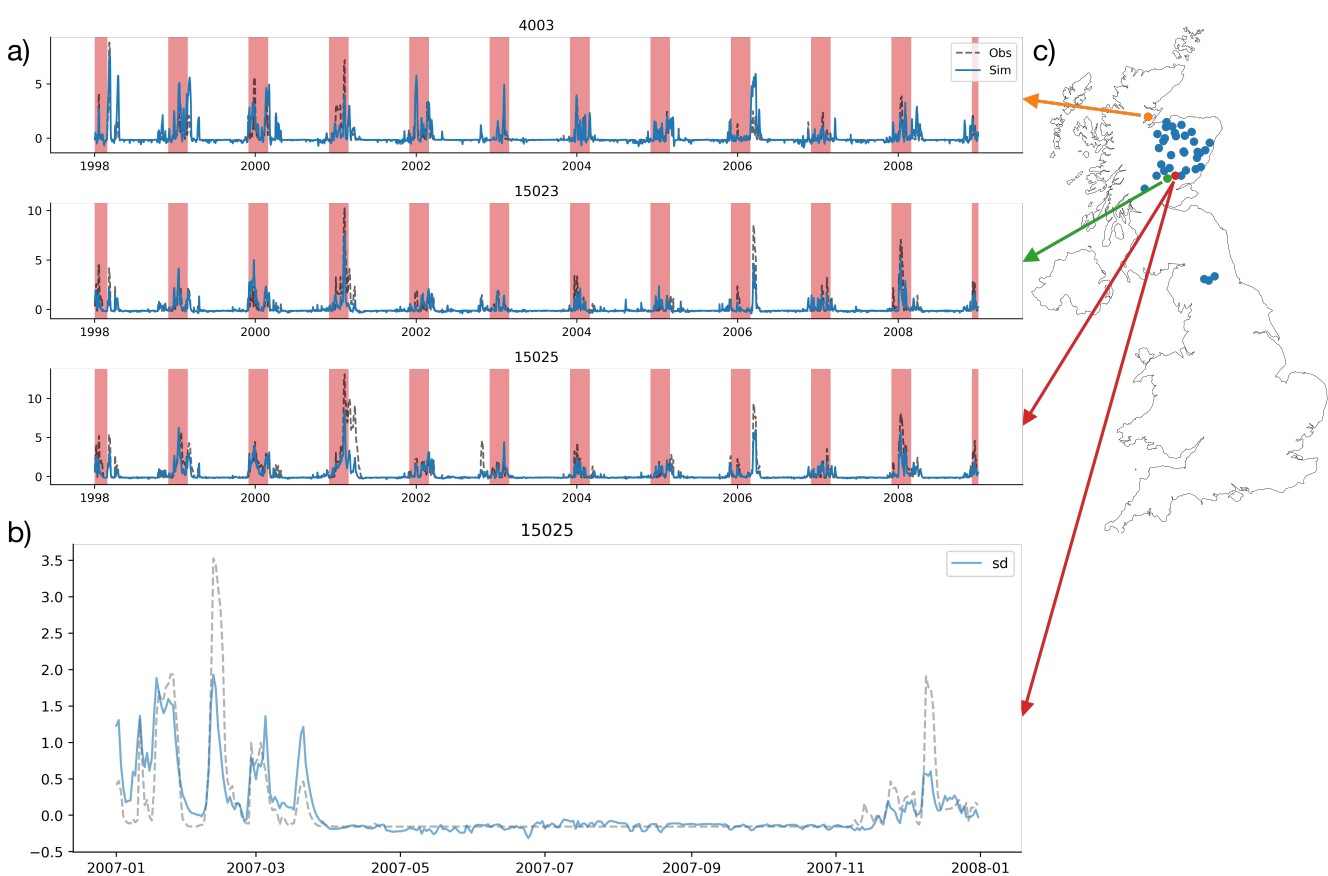

**Figure 5.** (a) The probe simulation (blue line) plotted against the snow depth variable from ERA5-Land (black dashed line) showing the correlation between the cell-state dimension and the target variable for all years in the training dataset (1998–2008). The red-shaded regions represent winter months (December, January, February). Each subplot shows results for a selection of the snowy catchments as selected from (c). (b) Probe predictions for the snow depth target variable for a single snowy station, 15025 in the Cairngorms (the red point in (c) ), for a single year 2007. (c) shows catchments with significant snow processes in Great Britain (defined as having a fraction of precipitation falling as snow greater than 5% using the CAMELS-GB frac_snow variable, which is the percentage of precipitation falling as snow) are concentrated in the Grampians range in North East Scotland and the Pennines in Northern England.





## 4 Discussion

### 4.1 The LSTM has learned physically realistic mappings

We find evidence that learned representations of catchment storages, as encoded by the LSTM state-vector, are predictive of a broad range of catchment hydrological storages. We have evidence that the LSTM, trained on a large dataset of 669 catchments,

learns to model soil moisture processes and snow water processes internally. We have tested these findings with different initial conditions by initialising the LSTM with a different random seed, with different probe designs and with different data products. We find that our results are robust to these different setups. This is despite the LSTM never having seen these data, nor being constrained to model these processes. It is worth emphasising that the LSTM is trained to predict only discharge, using information from three meteorological drivers: temperature, precipitation and potential evaporation; as well as 21 static

catchment attributes describing topographical conditions, climatological conditions, land cover types and soil texture (Lees et al. (2021) Table 2). No direct information about snow accumulation and ablation, nor soil moisture is included in the training data. The LSTM has to learn these processes itself from the raw meteorological inputs and the catchment attributes, determining that these concepts are useful in solving its training task, i.e. predicting discharge. The states of the LSTM are learned through backpropagation, and therefore, these learned states are deemed useful by the model in the training process

for minimising the error in the discharge signal. This finding offers evidence that the LSTM is learning a physically realistic mapping from inputs to outputs that corresponds with our understanding of the hydrological system. While there exists a large set of possible mappings from inputs to outputs, the LSTM converges on an answer that reflects our physical understanding. Although we only show results from one model initialisation here, these findings are robust to different initialisations (different random seeds).

We should not be entirely surprised by the finding that the LSTM has identified a physically realistic mapping from inputs to outputs. Neural networks are adept at learning the simplest solution to a given task. Since we are training the model to predict discharge in hundreds of basins across GB, the easiest solution is to learn the underlying physical relationships, since the alternative requires learning spurious correlations for all catchments. Nonetheless, by demonstrating that the LSTM learns physically realistic mappings we can imagine interesting opportunities for hydrologists to: (a) provide predictions of interme-

diate hydrological variables such as soil moisture and snow depth (b) explore what information remains in the cell-states that has not already been identified as important for soil moisture or snow processes.

   The former opportunity (a), that the probe analysis offers a means to predict intermediate variables, is interesting for two reasons. Firstly, the LSTM produces more accurate discharge simulations than any other hydrological model, and so we might expect that the intermediate variables are also better represented by the LSTM. Secondly, since the LSTM generalises to

unseen basins (Kratzert et al., 2019b; Frame et al., 2021), we might expect that the probe analysis also generalises to unseen basins. Exploring probe predictions in ungauged basins reflects an exciting area of future research. This could be done by learning the probe weights for certain basins, and then predicting on basins that have not been seen by the LSTM or probe before. The latter opportunity (b), that the LSTM allows us to learn something potentially "unknown", is an exciting aspect of using LSTM-based models for hydrological modelling. Exploring remaining values in the LSTM state-vector that have not





already been assigned to a concept offers one promising avenue for identifying important processes or variables unaccounted for in the predictors (e.g. water management); and/or explaining the performance difference between the LSTM and traditional hydrological models. Once identified, we have the opportunity to incorporate this information back into traditional hydrological models. Alternatively, we can employ feature importance metrics, such as the integrated gradients method, to identify the signals that are most informative, and then reason about what these signals might represent. Such approaches may, for example,

allow us to detect anthropogenic anomalies, such as reservoir operation decisions, changes in equipment, or biases in the observed data.

Since the LSTM is often the best performing rainfall-runoff model for discharge (Kratzert et al., 2018, 2019c; Gauch et al., 2021; Frame et al., 2021; Gauch et al., 2020), it makes sense to explore the soil moisture that the LSTM associates with a given level of discharge. Soil moisture is an important variable for understanding the runoff responses to precipitation events (Sklash

and Farvolden, 1979), for assessing drought stress (Manning et al., 2018), and for assessing the land-surface response to future climate change (Samaniego et al., 2018). We find that the LSTM is better able to model shallower layers of soil moisture than deeper layers (see Table 1). This likely reflects the fact that soil moisture in these shallower layers is more closely related to the discharge signal.

**Table 1.** Median catchment correlation scores over all 669 catchments for the training period for each soil water layer, 1 (0cm–7cm), 2 (7cm-29cm), 3 (29cm-100cm), 4 (100cm–289cm).

|  | Soil Layer 1 | Soil Layer 2 | Soil Layer 3 | Soil Layer 4 |
| --- | --- | --- | --- | --- |
| Median Correlation | 0.85 | 0.90 | 0.89 | 0.77 |

The probe analysis presented here represents one method for extracting intermediate hydrological concepts from the state

vector of the LSTM. Our purpose of using such a probe is to interpret how information is processed by the LSTM. We recognise that we should not necessarily expect to use the probe output for more than interpreting the internal mappings of the LSTM. However, it is interesting to consider what catchment soil moisture anomalies the LSTM learns to associate with given discharge outputs. Fig. 6 shows the soil moisture anomalies as predicted by the probe.

We tested the probe against satellite derived soil moisture observations, using the blended active and passive ESA CCI

Soil Moisture product to see if we get consistent results (Dorigo et al. (2017), see Appendix B). The results are similar to the results obtained when using the ERA5-Land data, with lower absolute correlation scores. Moreover, we see consistent performance declines when performing our two control experiments, i.e. shuffling the LSTM state vectors in space or shifting them in time (Appendix A), suggesting that soil information stored in the LSTM state vector is specific to each given catchment and timestep. We designed these control experiments to determine whether soil moisture time series can be reproduced from

unrelated inputs. The results give evidence that the extracted signals are specific and therefore, that the results are unlikely due to chance.

Using alternative products such as ESA CCI Soil Moisture (see Appendix B), a blended product combining multiple satellite-derived soil moisture estimates, comes with its own model assumptions and uncertainties. One key issue with remotely sensed



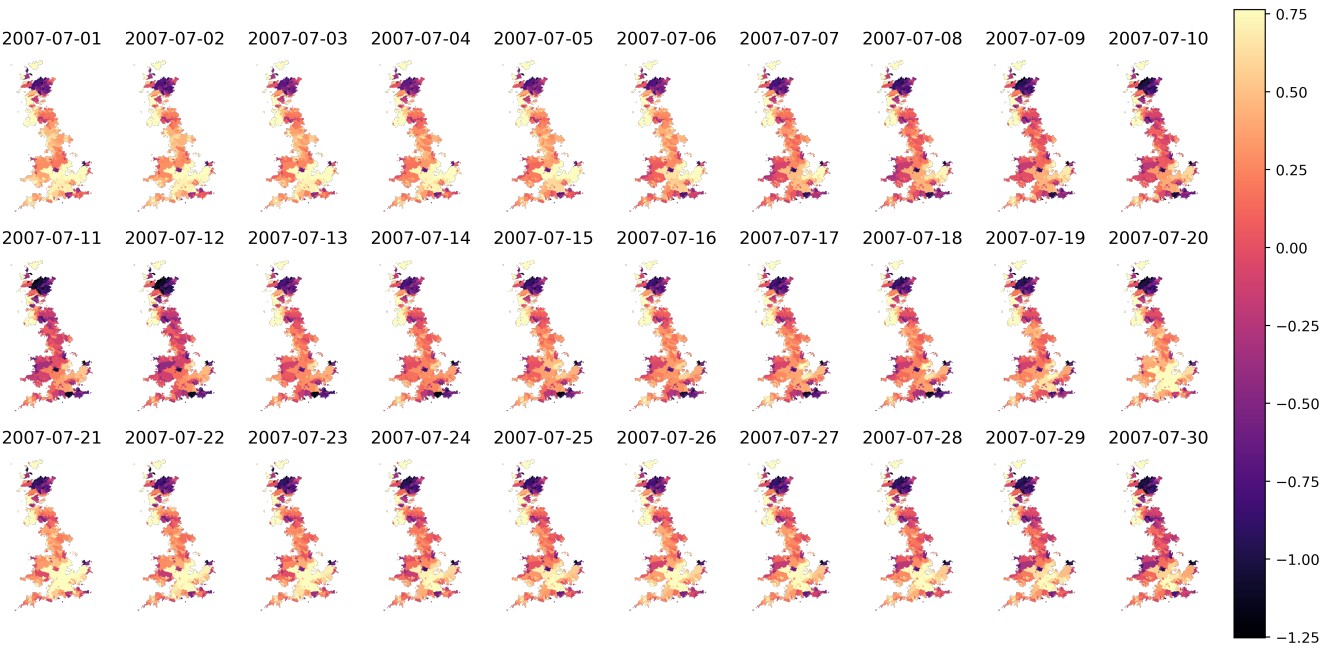

**Figure 6.** The lumped (catchment averaged) soil moisture for 0–7cm at 30 daily timesteps covering most of July 2007.

soil moisture is that the measured soil moisture is restricted to the top-layer (5–10cm), and the depth of the satellite derived

estimate is itself a function of the water content and surface roughness (Dorigo et al., 2017). Furthermore, it is difficult to

measure and define catchment-scale soil moisture in the real world, but we are running our experiments using data lumped at

the catchment scale. Neither reanalysis nor satellite-derived soil moisture reflects a true in-situ observation of catchment-scale

soil moisture. Our approach was instead to let the LSTM learn the most effective mapping from inputs to outputs, and we

interpret the intermediate state-vector, probing for the concepts that we expect to find given our expectations.

With respect to examining the learned representation of snow processes, one important aspect needs to be discussed. The

snow probe was trained on catchments with a fraction of precipitation falling as snow greater than 5%. Training a linear probe,

with a penalized squared error loss function (see Eq. 6), to predict snow depth on all 669 catchments caused the learned weights

of the probe to be zero and no signal was detected. This is because only 33 of the catchments have significant snow processes.

In contrast, when we trained the probe on catchments where we knew snow processes were occurring, we found that the snow

processes were well captured by the variability in the cell state values. Note that the probe is still trained on the same LSTM.

If we look at the cell-state values that had the largest weights in the linear probe, we can see that there exist snow-like signals

which the probe incorporated into its representation of snow depth (Figure 7).

Our experiments outlined above explore whether known hydrological concepts (i.e. water stores) are captured by the LSTM.

Our work here is the first step towards interpreting the internal dynamics of LSTM based models. Future work could consider

how we might use feature attribution approaches (such as the integrated gradients method) to identify the most informative

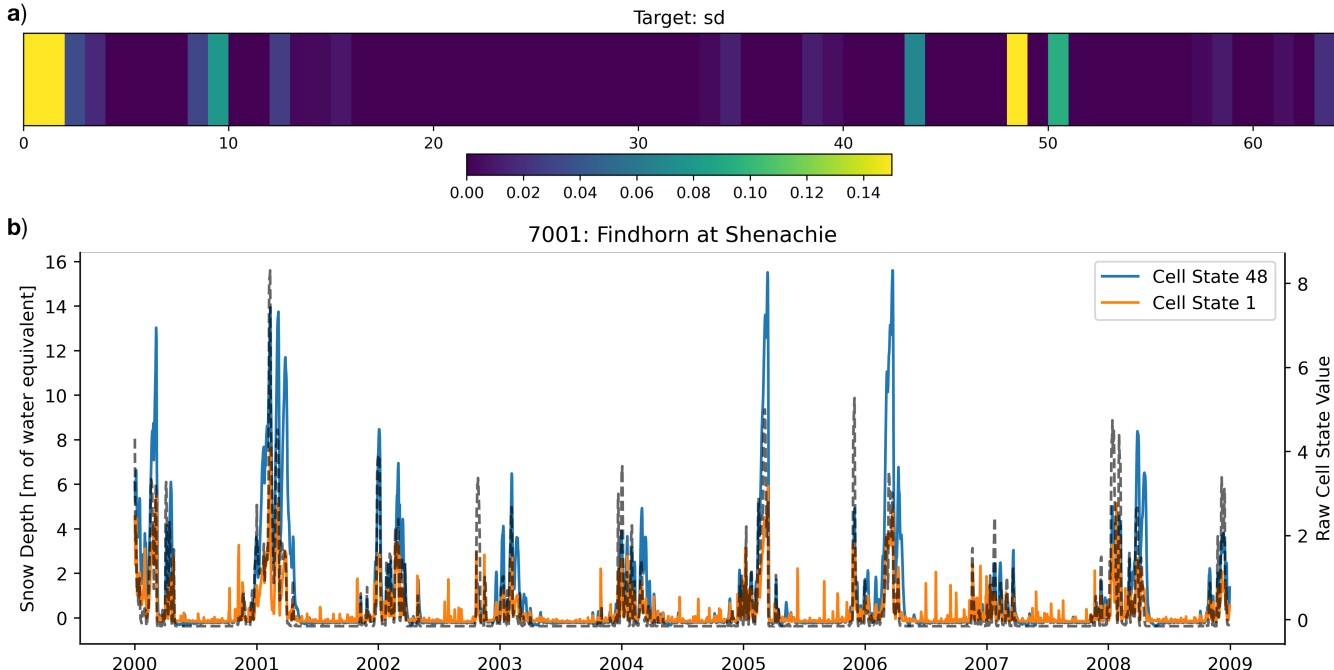

**Figure 7.** (a) We visualise the most informative weights where lighter colours means larger weights assigned to that cell state dimension by the linear probe. (b) The largest 2 dimensions, in terms of their assigned weight, from the above plot are then over-plotted on the snow depth target timeseries (grey dashed line), showing that they really do contain information about the winter snow signal, particularly Cell State 48 which varies very little in the summer months.

cell state values, or else examine the information contained within the states that do not correlate with snow or soil moisture processes.

## 4.2 The Catchment Biases in the Linear Probe

The probe predictions effectively model the dynamics of the soil moisture signals, however, they fail to predict absolute soil moisture levels, since they struggle to reproduce the catchment specific offsets that describe the mean catchment soil water volume. Two aspects of the training method for the probe need to be noted here. Firstly, we are training one linear probe for all catchments. This means that we have a single set of weights that linearly map from cell-state to each soil moisture level in all of the basins. We tested training one probe on each catchment, which caused the offset to be well modelled by the unique bias for each probe (not shown). Thus the offset problem is not an issue when we train one probe on each catchment. The reason that we chose to use a single probe for all catchments was that we expect the learned concept of the soil and snow processes to be invariant over space, i.e. that the relationships between discharge, precipitation, and intermediate hydrological stores (snow and soil moisture) are encoded in the state-vector in a way that can be easily extracted by our probe.



In order to train the linear probe, we calculate a global normalisation, such that each catchment soil moisture is a value relative to the GB-wide mean. Since we are training a linear model, we should expect the observed behaviour of our probe, which captures the correlation and dynamics of the signal well, but learns a single "intercept" or bias-term for all catchments. This is because the optimisation of a linear model is a convex optimisation problem, where there is one global minimum. The model has a single intercept (bias) parameter for all catchments (James et al., 2013). The chosen bias term is the one that minimises the residual sum of squares, and this is the mean of the training target data (the probe target, such as soil moisture volume level 1). In Appendix Sect C we explore how incorporating a catchment specific intercept allows the linear model to accurately model the mean catchment specific offsets. We do this by augmenting the state vector with a one-hot encoding of gauge IDs. Furthermore, a non-linear probe is also able to more effectively model mean catchment soil saturation conditions (Appendix Sect E). Therefore, the information can be extracted from the LSTM, however, a linear probe is not sufficiently powerful to do so. That being said, the changes in relative soil saturation levels are clearly well modelled as demonstrated by Fig. 3 and the high correlation scores in Table 1 and Fig. 2.

### 4.3 Probes offer a means of interpreting the learned representation of the LSTM

The probe analysis that we have undertaken here provides a means of interpreting the internal states of the LSTM. We are interested in extracting the information content of the LSTM state-vector, allowing for the fact that this information may be stored across multiple cell state values.

There is no reason why the LSTM should store that information in a single cell, rather than distribute information across multiple cells. In fact, the LSTM could also model a process as the difference of two (or more cells) or any other combination of multiple different cells. Indeed it is likely that information is distributed due to the process of dropout. Using dropout randomly sets certain weights to zero during training in order to prevent the neural network overfitting, preventing the co-adaptation of weights such that one weight "corrects" the influence (Srivastava et al., 2014). However, this also means that the network can potentially learn the same process in two different places in the network, since the network must be robust to those weights being "switched off" when dropped out.

The benefits of using the linear probe are twofold. The first is that our probe is relatively inflexible, and therefore, we can be confident that the probe is not overfitting to the targets (Hewitt and Liang, 2019). Indeed, we perform a number of further experiments to check for spurious correlations, as described in Appendix Sect. A. Our experiments demonstrate that the findings show the information from the LSTM state-vector is both catchment and time-specific, i.e. that the correlations are significantly better than a sensible benchmark, ensuring results are robust and unlikely due to spurious correlations. This is an important result because it ensures that we are finding meaningful correlations between the information in the LSTM state-vector and the soil moisture target variable. The second benefit of using the linear probe is that we can easily interpret the probe weights. However, there are limitations of this approach too. The linear probe limits the obtainable information to certain forms of information storage. In theory, it is possible for the LSTM to use information linearly for certain flow-regimes, and then use it differently for other flow regimes, even within a single basin. Thus, it is possible that the linear probe is unable to extract relevant information from the cell states that are being used by the LSTM. To test this we could use a fully connected



neural network or any other non-linear regression model (see Appendix Sect. E). However, using a more complex model as a probe also comes with its own downsides. We lose the interpretability that comes from inspecting the probe weights and also increase the chance of getting false-positive results (Hewitt and Liang, 2019).

An open question remains, what other information is captured by the state-vector values that are not already assigned to a particular hydrological concept? Exploring these remaining cell states offers the potential to identify the underlying reason for the increased performance of the LSTM. Given that soil moisture and snow processes are already included in most hydrological models, the improvement of the LSTM over the conceptual and process-based models is unlikely to be a result of these processes we explore here. It remains possible that the LSTM is better able to learn the complex interaction of these processes
with catchment-specific information, however, it is also possible that these remaining cell states contain information that describes other processes such as anthropogenic impacts on the hydrograph including withdrawals, transfers and reservoir management rules.

## 5   Conclusions

LSTM-based rainfall-runoff models offer good hydrological performance, however, interpretation and exploration of the con-
cepts and structures that these models have learned is still in its infancy. In this paper we have explored the information captured by the LSTM state vector using tools from machine learning interpretability research.

We use a linear probe to map the state-vector onto a target variable, and find that there is sufficient information in the state-vector to represent the temporal dynamics in both soil moisture (at different levels) and snow depth from commonly applied data products. The state-vector of the underlying LSTM is trained only on meteorological forcings, static catchment attributes
and asked to predict discharge.

Ultimately, these results suggest two key conclusions. First, the LSTM is learning a physically realistic mapping from me-teorological inputs to discharge outputs. The concept of a soil store and snowpack is encoded in most conceptual hydrological models (Beven, 2011). We therefore have evidence that in the UK the LSTM is learning to get the right results for physically-plausible reasons.

Finally, the conceptual approach that this paper has taken, using a linear probe, offers an effective method for extracting information from LSTM state vectors. Wherever LSTMs are applied, there is the possibility of exploring $c_t$ in a similar way. This method has been applied in natural language processing, but as far as we are aware there are no applications of this method to LSTMs in Earth systems sciences.

*Code and data availability.*   CAMELS-GB data is available at: https://catalogue.ceh.ac.uk/documents/8344e4f3-d2ea-44f5-8afa-86d2987543a9.
The FUSE benchmark model simulations are available at: https://data.bris.ac.uk/data/dataset/3ma509dlakcf720aw8x82aq4tm. The neural-hydrology package is available on github here: https://github.com/neuralhydrology/neuralhydrology. The exact code and notebooks used to generate all of the plots can be found here: https://github.com/tommylees112/neuralhydrology/tree/pixel. The models, probe predictions and config files can be found here: https://doi.org/10.5281/zenodo.5600851



## Appendix A:  Control Experiments

We want to be sure that the signals found by the probe are specific to a given catchment and time, and control for false positives (finding a strong correlation between the LSTM state variable and the intermediate target variable of interest). Testing for the probe confounder problem, which describes "when the probe is able to detect and combine disparate signals, some of which are unrelated to the property we care about" (Hewitt and Liang, 2019) requires that we ensure that our detected signals are specific to the catchments (spatial specificity) and times (temporal specificity). Our hypothesis is that the LSTM is learning information

that is specific to soil-moisture and snow processes for a given catchment at a given time. In order to test this hypothesis, we designed two control experiments. The first involves spatial shuffling, where we take the LSTM state-vector from Catchment A and test whether the information content is sufficient to accurately model Soil Moisture in Catchment B. Spatial shuffling tests for spatial specificity and can be seen in Figure A1.

The second experiment shifted the data in time, testing for temporal specificity. We tested the performance of the probe with

inputs shifted by 180 days, breaking the temporal link between inputs and outputs (Fig. A2).





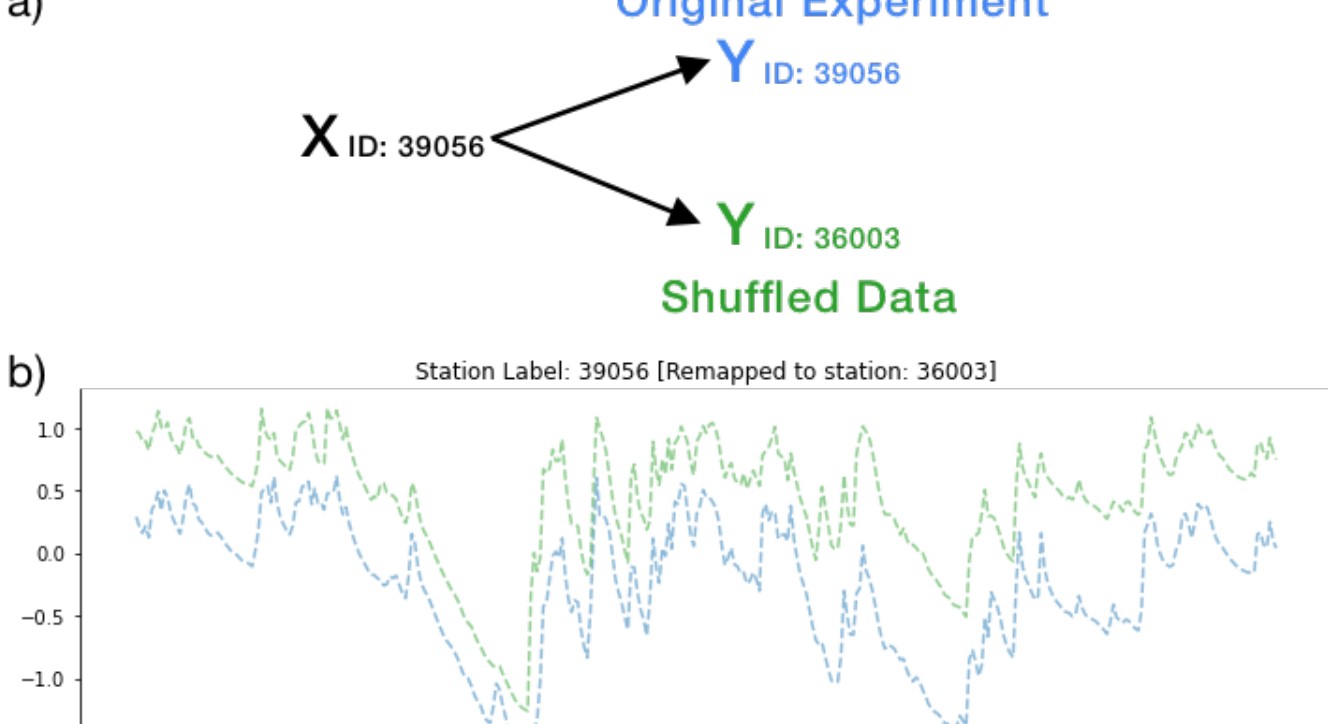

**Figure A1.** (a) Conceptual diagram explaining spatial shuffling. In the original experiment, $c_t$ from Gauge ID 39056 is linked to the time-series for that catchment 39056. The shuffled target instead asks the probe to detect the target variable time-series (belonging to Gauge ID 36003) using the $c_t$ vector from Gauge ID 39056. (b) Example time series from shuffled basins, where station 39056 is the original experiment and station 36003 is the shuffled experiment. It is worth noting that the soil moisture measurements are highly correlated in space (shown by the blue and green lines following each other), and in some instances multiple basins may fall within the same ERA5-Land pixel, in which case the information content will be the same.





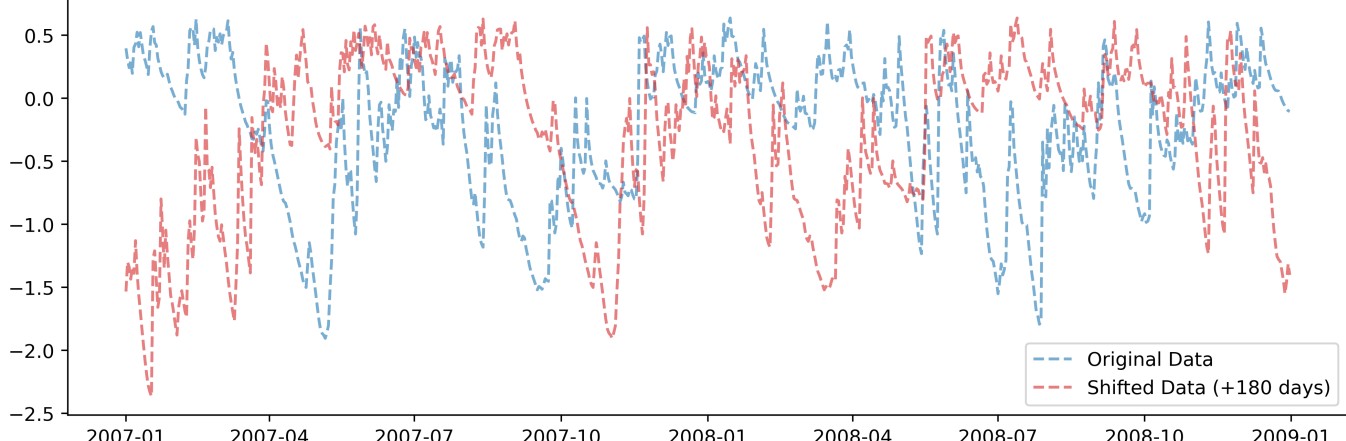

**Figure A2.** Shifting the input data by 180 days in time breaks the temporal link between the inputs (shown here) and the target variables (not shown). We fit the probe on the shifted data to determine how much less information it contains when compared with the original data.

The results show that the probe performances declined for both experiments, shifting the inputs in time and shuffling the inputs in space (Figure A3. A4). This suggests that the information captured by the LSTM state-vector is specific to the catchment and time. The original experiments had correlation scores of: 0.88 for swvl1; 0.90 for swvl2; 0.90 for swvl3; and 0.84 for swvl4 respectively. When shifting in space, these declined to 0.72, 0.76, 0.76, 0.68 for each target variable. When shifting in time, these declined to 0.39, 0.26, 0.63, 0.50. It is likely that the larger performance drop for shifting in time is because of the high degree of spatial correlation in the catchment-averaged soil moisture time series. Ultimately, these experiments give us a high degree of confidence that the observed correlations are unlikely due to chance, and that the information that the probe is extracting from the LSTM state vector reflects the hydrological variables we compare against.



**Figure A3.** Shuffling in space. (a) Histograms of the original experiments with unshuffled data, repeated from Fig. 2. (b) Histograms of catchment correlation scores after having trained the probes on data shuffled in space. The performances decline quite significantly when compared with the original experiments.

**Figure A4.** Shifting in time. (a) Histograms of the original experiments with unshuffled data, repeated from Fig. 2. (b) Histograms of catchment correlation scores after having trained the probes on data shifted in time.





**Appendix B: Probing the ESA CCI Soil Moisture**

We also tested using the ESA CCI Soil Moisture as our target variable (Dorigo et al., 2017; Gruber et al., 2019). For this analysis
we use the blended product combining active and passive satellite-based sensors (the combined product). The data is a globally
available long-term daily satellite soil moisture product that covers the period from 1978–2015 at a 0.25° resolution (Dorigo
et al., 2017). The daily estimate is noisy and therefore, we smoothed the daily data using a 7-day moving average window.
The product is of lower spatial and temporal resolution than ERA5-Land, however, it provides an independent estimate of

soil moisture based on satellite-derived soil moisture estimates. Neither product is an in situ observation and so both rely on
modelled assumptions.

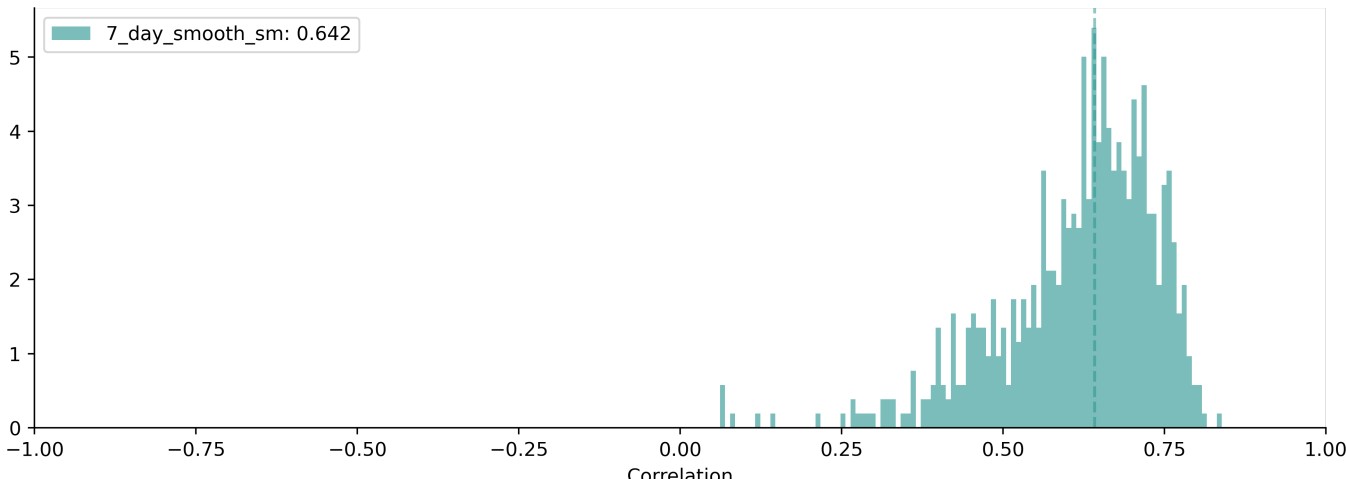

**Figure B1.** Histogram of catchment correlation scores for the probe-simulated soil moisture and the 7 day smoothed ESA-CCI Soil Moisture.
The median score is 0.64.

The probe captures the temporal dynamics of the soil moisture signals, but struggles to reproduce the catchment specific
variability, failing to match the peaks and troughs associated with the target variable (ESA CCI soil moisture). The overall
results indicate that the LSTM probes are less able to reproduce ESA CCI Soil Moisture than the signals found in ERA5-Land.

Ultimately, we chose to use the ERA5-Land results because of the higher spatial resolution, and therefore, increased specificity
of catchment averaged soil moisture values. We can see the intercomparison of these factors in Fig. B4, and Fig. B4.

**B1   How Similar are the ESA CCI Soil Moisture and the ERA5-Land Soil Moisture?**

We show the correlation between catchment-averaged soil moisture from the two products in Fig. B3. There is a median
Pearson correlation of 0.50 between the products. The underlying spatial resolution of the two products, and the median spatial

patterns can be seen in Fig. B4. This demonstrates that the ESA CCI soil moisture and the ERA5-Land soil water volume level
1 have similar spatial patterns, with low soil saturation in the Scottish Highlands, saturated soils on the Scottish West Coast and







**Figure B2.** Time series of probe predictions (coloured lines) compared with the target variables (grey dotted lines). We show two catchments here, 54018 and 15021 for the ESA CCI Soil Moisture Products which is the same as Fig. 3





relatively unsaturated soils in Central and Eastern England. However, the spatial resolution for ERA5-Land is much higher, with 6 times as many pixels as the ESA CCI data.

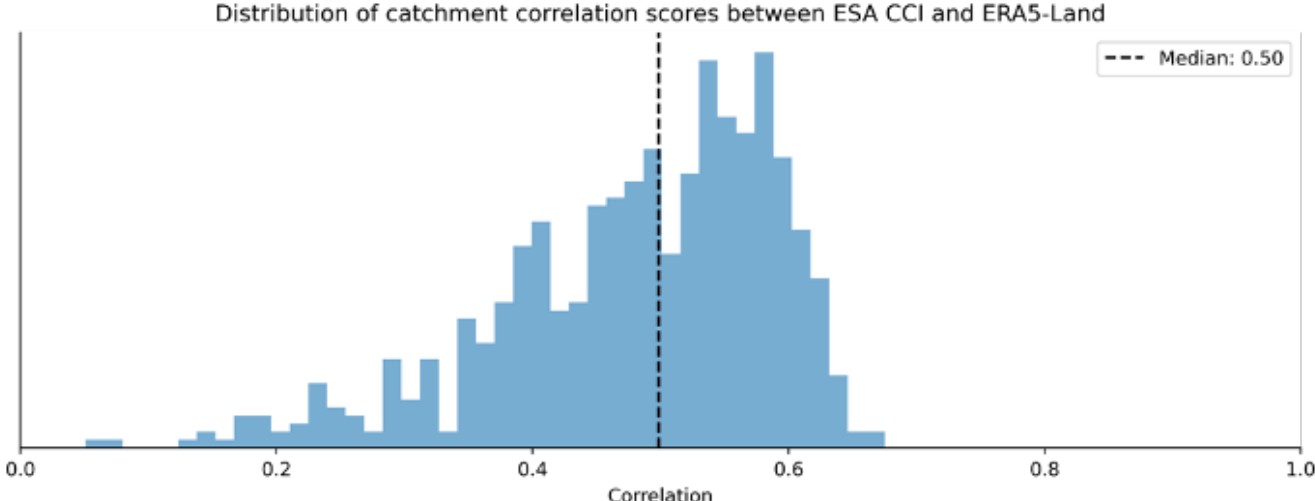

**Figure B3.** Histogram showing the distribution of correlations between ESA CCI Soil Moisture and the ERA5-Land Soil Moisture Products (Soil Water Volume 1 - 0–7cm) for each catchment.



**Figure B4.** The spatial resolution of the two soil moisture products (ESA CCI Combined left, ERA5-Land Soil Water Volume Level 1 right). The ERA5-Land data (87 x 93) contains roughly 6 times as many pixels as the ESA CCI soil moisture (35 x 37).





**Appendix C: Investigating the Catchment Specific Probe Offsets**

As discussed in Sect. 4.2, a simple linear model is not able to predict catchment-specific offsets because it is constrained to model a single bias term for each catchment. In order to minimise the residual sum of squares, the optimum solution is the mean of the target data in the training period. Given that we normalize the target data, this global mean is equal to zero. Therefore, the learned bias in the linear probe will be zero and we cannot expect the probe to reproduce catchment specific offsets. One simple solution to this problem is to include one hot encoded information for each catchment as input features to the regression.

We augment our input state dimensions (64) with a binary encoding representing which catchment that data point is drawn from (Fig. X). Then the linear model can learn a catchment-specific bias, which will be the mean value for that catchment.

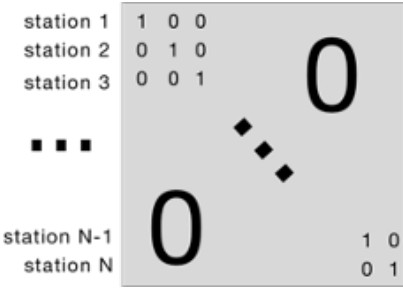

**Figure C1.** A diagrammatic explanation of the one hot encoding vectors, where each station is given a unique encoding by using an identity matrix of size $N$ (number of stations). Each gauging station is encoded by a vector of 1s and zeros (a vector of length $N$). We then append this vector of OHE data ($X_i^{OHE}$) to each state vector ($c_i$) to create an augmented input to the linear probe, $X_i^*$.

We train a linear probe ($f_\beta$) using augmented input data ($X_i^*$), combining the state vector from the LSTM ($c_i$) with an encoding of the gauging station number ($X_i^{OHE}$). This is then included as the input to the probe to produce a predicted output ($\hat{s}_i$).

$$X_i^* = [c_i, X_i^{OHE}] \tag{C1}$$

$$\hat{s}_i = f_\beta(X_i^*) \tag{C2}$$

In Fig. C2 we can see that including the augmented input variables hugely reduces the biases in the probe outputs, and largely solves the catchment offsets, as expected. This result is expected because the linear probe now can learn a catchment-specific intercept term (a catchment specific bias) that adjusts the probe outputs to the appropriate mean saturation conditions.

If we observe these patterns over all catchments we can see that the performance improvement for different soil moisture levels is marked for RMSE (Fig. C3a) but is small for correlation (Fig. C3b). This is because the modelled dynamics are very similar (high correlations), but accurately predicting the mean catchment saturation at different soil levels greatly reduces the absolute squared error.





**Figure C2.** Time series of probe predictions with augmented inputs, including one-hot encodings of the gauge station ID (coloured lines) compared with the target variables (grey dotted lines). We show two catchments here, 52010 - Rea Brook at Hookagate and 15021 - Burn at Burnham Overy. The results show that we learn a catchment specific weight for adjusting the predictions and address the biases for each catchment.

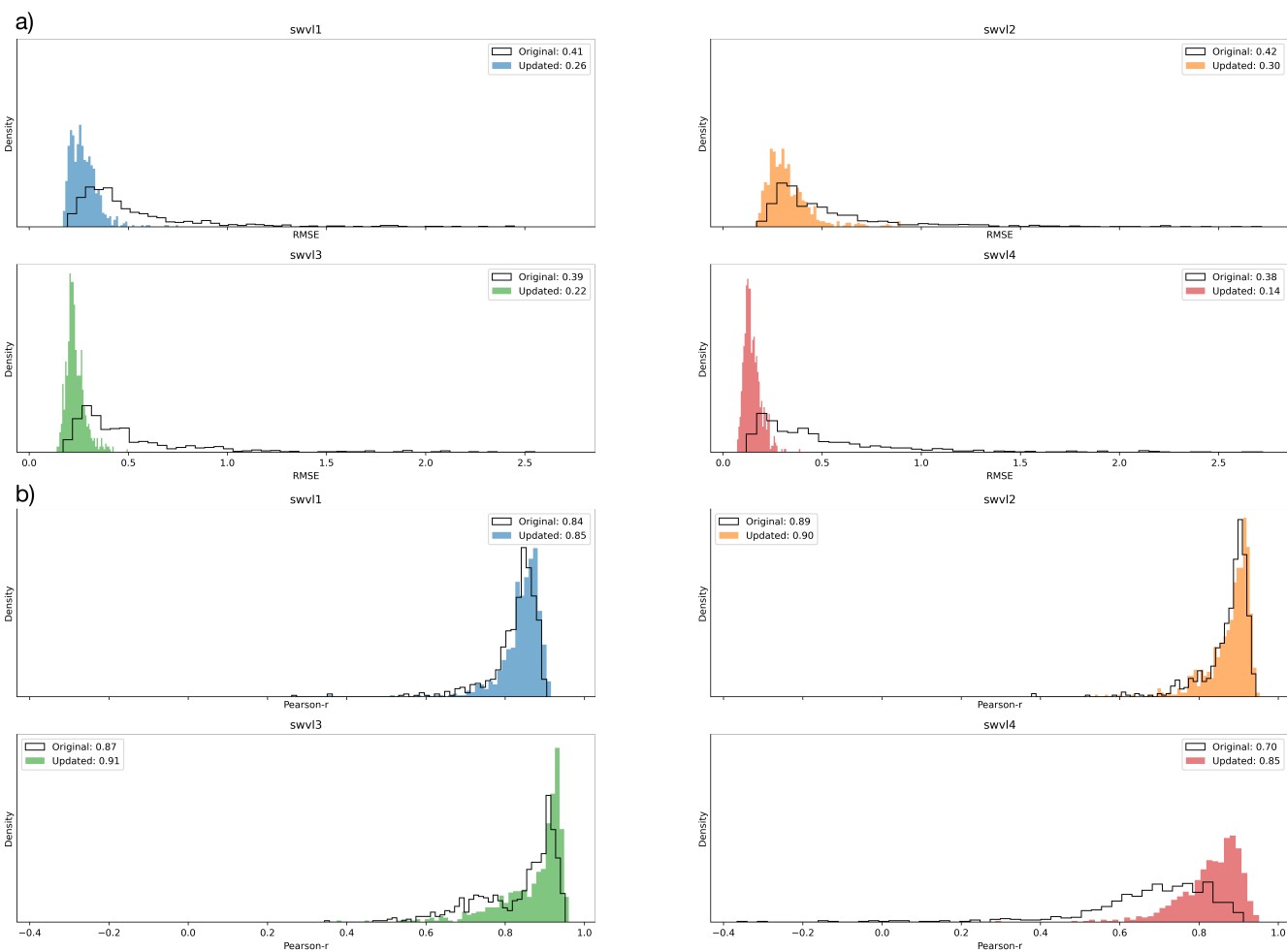

**Figure C3.** Histograms showing the distribution of catchment error metrics. The original linear regression model is shown as a solid black line. The coloured histograms correspond to the updated model with augmented inputs (using one hot encoded data) for each soil water level. Subplot (a) contains histograms showing the root mean squared error (RMSE) metric is much improved when modelling with the one hot encoded data. Subplot (b) contains histograms showing the correlation metric, showing that the dynamics are well modelled by the original linear regression, since the different biases do not influence the correlation metric.



## Appendix D: Spatial Context of Demonstration Basins

In the main body of text we have shown soil moisture timeseries for two catchments that were selected to demonstrate a well-performing catchment and a catchment with a significant bias to demonstrate the problem with modelling catchment offsets. They were randomly chosen from the lower tercile of the RMSE error distribution and the upper tercile. The size, wetness and elevation of these two catchments, Gauge 54018: Rea Brook at Hookagate and Gauge 15021: Lunan Burn at Mill Bank, are described by Fig. D1.

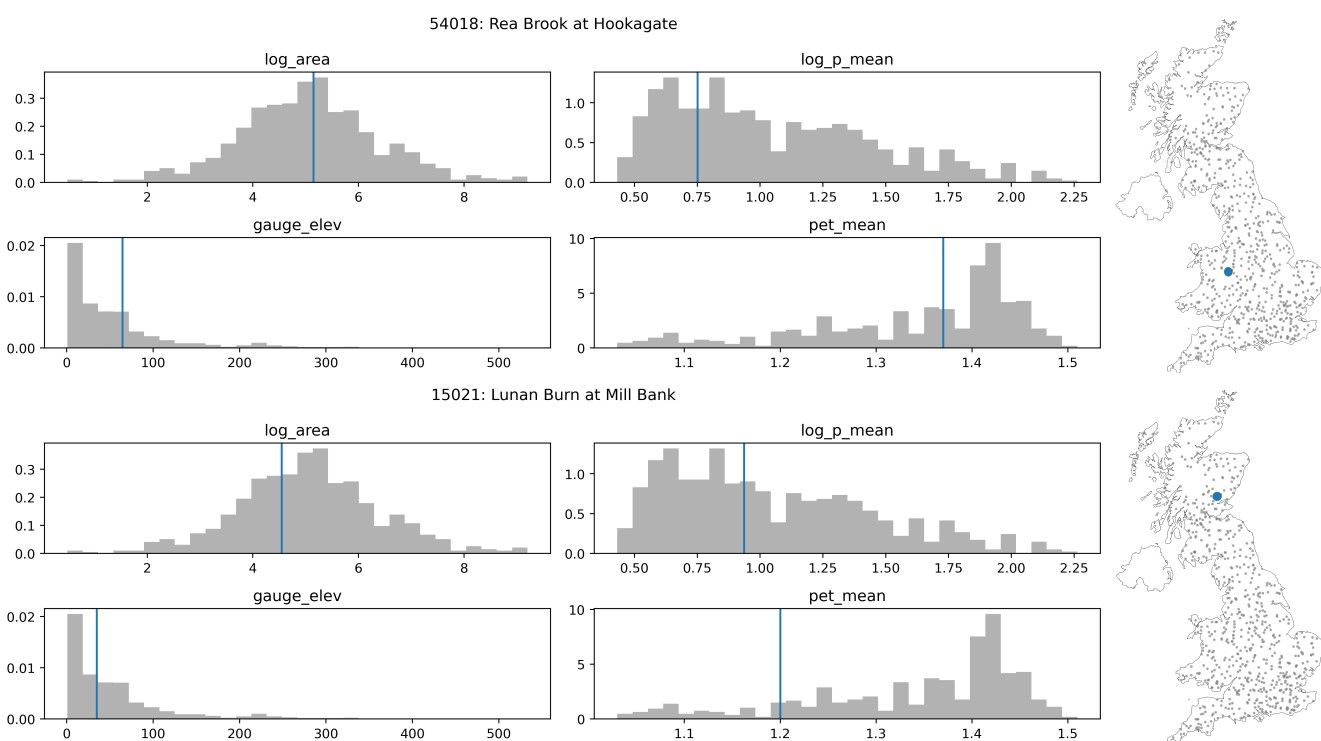

**Figure D1.** The spatial context of the two demonstration catchments shown in the Figures in the sections above. For both Gauge ID: 54018, Rea Brook at Hookagate (a), and Gauge ID: 15021, Lunan Burn at Mill Bank (b), we show the log area, the log mean precipitation, the gauge elevation and the mean potential evapotranspiration, as well as the location of the station on the map on the right.

## Appendix E: Non Linear Probe Results

We initially explored a linear probe for the simplicity of interpretation. However, there is no reason why we could not also use a non-linear model to extract non-linear patterns from the cell states. We tested a simple two layer neural network. We trained a fully connected network with hidden sizes of 20 and 10 for the first and second layer respectively. These layer sizes were chosen using a grid search algorithm, where we determined that this was the optimum layer size for minimizing the loss

function (RMSE). We used the Rectified Linear Unit (ReLU) as our activation function which was chosen because it is known to reduce the risk of vanishing gradients (Nair and Hinton, 2010), and trained our probe using stochastic gradient descent with the Adam optimizer (Kingma and Ba, 2014).

    The correlation scores are similar, as shown by the overall fit between the predicted time series and the observed timeseries in Fig. E1. The non-linear probe, unlike the linear probe, is able to model catchment specific offsets most obvious when looking

at catchment 15021 and soil water volume level 4 (lower figure, lower-right subplot Fig. E1). This pattern is repeated across other catchments, and the offset problem is much reduced for the non-linear model. This suggests that while the information may not be linearly extracted, a non-linear combination of the cell states does contain the information for the catchment specific offsets. Further research will consider various probe architectures and the other hypotheses outlined above to more fully explore whether catchment offset information is captured by the LSTM state-vector.

*Author contributions.* TL produced all experiments, plots and code. The manuscript was reviewed and edited by all co-authors. SD and SR provided overall supervision on the manuscript from a hydrological and machine learning perspective respectively. SD, SR, DK, FK and MG helped to conceptualize the research questions and design the experiments. All co authors contributed to the overall idea of exploring LSTM-based interpretability. SR, FK, DK, RK, JDB and MG provided machine learning inputs. DK, FK and MG came up with the idea for the sanity-check experiments, when data is shifted in time and shuffled in space. SD, LS, JDB, and PG provided hydrological inputs and

direction.

*Competing interests.* We declare that no competing interests are present.

*Acknowledgements.* The authors would like to thank the teams responsible for releasing CAMELS GB (Coxon et al., 2020b) and the authors and maintainers of the neuralHydrology codebase for training machine learning models for rainfall-runoff modelling. TL is supported by the NPIF award NE/L002612/1; SD is supported by NERC grant NE/S017380/1. We further acknowledge support by Verbund AG for DK and

by the Linz Institute of Technology DeepFlood project for MG. Part of the research was developed in the Young Scientists Summer Program at the International Institute for Applied Systems Analysis, Laxenburg (Austria) with financial support from the United Kingdom National Member Organization.





**Figure E1.** Time series of non-linear probe predictions (coloured lines) compared with the target variables (grey dotted lines). We show two catchments here, 54018 and 15021 (the same as Fig. 3) and four soil moisture levels, swvl1 (blue), swvl2 (orange), swvl3 (green), swvl4 (red). The non-linear probe shows reduced probe offset errors, showing that the information is contained but the linear probe is by definition not able to model catchment-specific intercept terms to account for unique biases.





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
