# Peer review of "Hydrological Concept Formation inside Long Short-Term Memory (LSTM) networks"

_Hydrology and Earth System Sciences, 2021_

## Author Comment (AC2)

**Response to Reviewers for Manuscript: Hydrological Concept Formation inside Long Short-Term Memory (LSTM) networks**

Comments/Text of Referee posted in **black**, our text in **purple**.

**Reviewer 1: Lukas Gudmundsson**

Comments/Text of Referee posted in **black**, our text in **purple**.

The paper submitted by Lees et al. aims at advancing the interpretability of neural networks used for rainfall-runoff modelling, focussing in particular on Long Short Term Memory (LSTM) architectures. LSTMs (a special type of neural network) have in recent years been popularized for rainfall-runoff modelling. The resulting models perform very well but lack a stringent physical interpretation. An interesting property of LSTMs is that they contain internal states – similar to internal states (storages) in classical hydrological models. Lees et al use statistical "probes" to link these LSTM states to independent estimates of soil moisture and snow storage. The analysis shows that LSTM states mimic soil moisture and snow storage dynamics although these quantities were not used for model calibration.

This paper is to my knowledge the first systematic attempt to interpret the internals of an LSTM used for rainfall-runoff modelling in a systematic manner, although un-systematic examples have been emerging in the literature. The application of LSTMs for rainfall-runoff modelling is state of the art and the use of "probes" based on linear regression makes intuitively sense.

Beside this, I personally value the authors effort to assess the robustness of their results by (i) applying the "probes" to both re-analysis based and remote-sensing based soil moisture estimates and (ii) by randomization based statistical testing.

The objective of the study is clearly stated: I.e., to test if internal states of an LSTM used for rainfall-runoff modelling at many catchments at once can be linked to independent estimates of soil moisture and snow. The paper is well structured to meet this objective and the data and methods are chosen accordingly.

We thank the reviewer for the summary of the manuscript. We are pleased that you found the methods intuitive and the robustness experiments useful. By addressing the reviewers' suggestions, we have revised our manuscript, and we believe our manuscript has significantly improved

Nonetheless the study leaves some open questions which the authors may want to further explore:

An obvious limitation of the proposed approach is that it relies on independent estimates of soil-moisture and snow (or other variables). Therefore, it does not allow for a self-contained interpretation of LSTM states.

This is correct. We are aware of this limitation and had the following sentence in our description of methods:
"*Since these probes are trained in a supervised way, we are currently limited to looking for known hydrological processes. Trained in this way, probes cannot be used to extract unknown information, since we require a target variable to fit the probe. This means that in the present study we are not looking for new hydrological understanding, or seeking to uncover as-of-yet undiscovered hydrological patterns, but explicitly looking for known physical processes in the learned LSTM representation.*".

In order to address the point that our approach does not allow us to diagnose unknown processes without prior estimates of those processes (e.g. soil moisture time series), we need to consider the LSTM architecture more closely. Except for a perhaps innate bias towards simpler solutions, the LSTM is not restricted in how it stores information in the internal representation. That is, the trained LSTM has no intrinsic reason to encode its representation in a manner that is directly accessible to humans or can be directly mapped to environmental variables. A lot of information can be stored within the cell states and it is not trivial to reverse engineer how it is used. Using probing allows us to isolate different functional groups in the cell state and trace their behaviour, by considering the probe outputs and how closely they relate to our time series of interest. Alternative approaches that use unsupervised methods can, and should, be considered. We discuss one of these approaches (principal components analysis - PCA) below.

We will also include a sentence in our Conclusion in order to make it clearer that this is a limitation of our approach:
"*One obvious limitation of this approach is that we rely on pre-existing estimates of our concepts of interest. Therefore, as proposed, this method cannot be used to identify unknown processes, due to the fact that we need a target variable to fit the probe*".

It would be interesting to know how much of the variance of the internal state vectors is captured in the resulting soil moisture estimates. How many independent signals are present in the 64 states? (e.g. estimated as the number of dominant principal components)

We thank the reviewer for this comment. It is an interesting line of enquiry and we present some preliminary analysis here.  If we have interpreted the reviewer correctly there are two aspects:
- How important is the soil moisture signal for discharge estimates?
- How many signals are present in the cell states?

In conclusion to the relatively long response that follows: we believe that probes with the elastic weights are the closest we have come to answering these questions thus far, since we try to extract the soil moisture signal from the least amount of cell-states in a linear (*naive*) way. Even with this approach we miss potential information, because we do not want to risk overfitting or loss of interpretability of the output model.

a) How important is the soil moisture signal for discharge estimates?
This is an interesting question and one that we spoke about in the discussion of our manuscript in two locations outlined below:
L278: "*Alternatively, we can employ feature importance metrics, such as the integrated gradients method, to identify the signals that are most informative, and then reason about what these signals might represent.*"

L319-322: "*Future work could consider how we might use feature attribution approaches (such as the integrated gradients method) to identify the most informative cell state values, or else examine the information contained within the states that do not correlate with snow or soil moisture processes.*"

We agree that this is an important component of cell state interpretability, and there has been work done by some of the authorship team that have explored this using the integrated gradient method discussed above (Kratzert et al 2020).  This is a different approach to the one outlined by the reviewer, that uses the automatic differentiation capabilities of the Pytorch LSTM to measure the sensitivity of the output with respect to each input feature. Ultimately, we chose to focus the paper on extracting intermediate signals of snow and soil moisture processes.

Ultimately, it is worth noting that the LSTM is not forced to learn anything about soil moisture or snow processes. The very fact that these signals exist in the cell state suggests that a representation of soil moisture was useful for the LSTM when modelling discharge.

b) How many signals are present in the cell states?
The principal aim of this paper was to examine the internal functioning of the LSTM and determine if the LSTM learns a physically realistic mapping from inputs to outputs. Unsupervised methods could be an alternative approach to the probe analysis for detecting signals contained in the LSTM cell state. There is additional work to be done in this area and we leave this for future papers, however, we present an initial exploration of this approach here.

PCA represents an unsupervised method for reducing a multidimensional dataset into "principal components" which are linear combinations of the original features that capture the most variance. It is a form of dimensionality reduction, since we can distil our 64 cell state dimensions into *n* principal components. We complete a preliminary analysis of the cell states using PCA below.

[Figure]

Figure 1: The ratio of variance explained by the dominant principal components, shown as a raw total (blue bar) and as a cumulative sum (blue line).

Analysis of the percent of variance explained by the leading principal components (Figure 1) shows that there is no clear cutoff defining the number of "dominant principal components". We could set a threshold for the amount of variance that must be described (e.g. 80% would leave 22 principal components), but the choice of this threshold is not clear a priori. That being said, we have here an estimate of the number of independent signals in the cell state, and it appears to be significantly less than the chosen hidden size of 64. This is expected, because the dropout regularisation during LSTM training incentivises redundant representations. A word of caution however, since PCA assumes variance as the proxy for "importance", yet the LSTM can easily learn a switching process that communicates which cell state is worth listening to in a particular hydrological context.

[Figure]

Figure 2: Visualising the 4 principal components that explain the most variance (coloured), plotted over the top of the discharge time series (black dotted line) for a single basin (10002).

In Figure 2 we observe the four principal components that explain the most variance in the original cell states matrix. In the previous comment, it was mentioned that a limitation of our *supervised* approach is that we require a target timeseries to extract these signals from the cell states. This is true, however, it is hard to see how we can escape this even after using an unsupervised method. Once we have extracted these linearly independent signals, we still require a target timeseries to determine whether our signal corresponds to the concepts we are looking for. This could be done, for example, by finding the principal components with the largest correlations to our intermediate storage variables, which is very similar to the probe approach taken in the manuscript.

A point that we want to emphasise is that the LSTM is not restricted to store linearly separable information in the cell state, it is itself a nonlinear regression model. The cell states can interact with one another and can display small variance but still be important for the LSTM computations. This makes using a method like principal component analysis difficult since the underlying state vector that we use as input to the PCA analysis violates the assumptions of the method (such as the assumption that variance describes the important axes of information).

We want to caution that not all of the processes captured by the LSTM cell state can be directly related to hydrological concepts, for example, in our experiments we have observed that the LSTM often contains something like a counter, a linearly increasing value that counts the number of timesteps from the first input time. This is useful for the LSTM computation but is not interpretable as a physically meaningful hydrological signature.

PCA is one of many methods in blind signal separation (closely related to *unsupervised learning*). Other established methods that would warrant further exploration include independent component analysis (ICA), non-negative matrix factorization and SVD. Ultimately, the use of unsupervised methods for distilling the LSTM cell state into human-interpretable signals is an interesting research question worthy of analysis in future publications.

**Minor issues:**

Inconsistency: Equation 1-2 use i_t as input. Equations 3-4 use x_t
We will update this as proposed. (Thanks!)

The paper somehow requires that the reader is familiar with how LSTMs work. I acknowledge the authors choice not to repeat the LSTM definition, also since it is available in many other publications. Nonetheless, this made it a bit more difficult to fully understand the paper.

We do understand this concern, and it can indeed be very difficult to find the balance between providing the right amount of information and repeating information from elsewhere. We thus added the following sentences for readers that are not too familiar with the LSTM:

We will update our manuscript to include a sentence that reads: "*For a more detailed description of the LSTM we refer to \citet{kratzert2019_ealstm}, particularly Figure 1 and Equations 1–12, both found in Section 2 Methods.*"

Furthermore, the description outlined L90-L109 gives an overview of the salient aspects of the LSTM for a hydrologist, making it clear that the state of the LSTM is conceptually similar to state variables in other hydrological models.

Elastic net: I assume that the description of the elastic net regularisation might be quite cryptic to readers who are not familiar with this tool (I am).
We agree and propose to update our explanation to a more informative explanation:

"*Our linear model, $f_{\beta}$, is a penalised linear regression model. We use the elastic-net regularisation that combines the $\ell_1$ penalty of lasso regression with the $\ell_2$ penalty of ridge regression. The reason for choosing the elastic-net regularisation is that when we have correlated features in $c_t$, The lasso ($\ell_1$ penalty) shrinks non-informative weights to zero, whereas the ridge ($\ell_2$ penalty) shrinks weights for correlated variables towards each other, thereby averaging these correlated variables \citep{friedman2010}. The elastic net is a compromise between these two penalties, where a higher $\alpha$ parameter encourages a sparser regression equation, and a lower $\alpha$ parameter encourages averaging correlated features \citep{friedman2010}.*"

Also: Given the large number of samples (# catchments x # time steps) and the relatively low number of predictors I wondered if a linear regression would perform equally well.

We tested the linear regression fit using ordinary least squares (specifically using scipy.linalg.lstsq via the sklearn.linear_model.LinearRegression) and the performances were poor. The highly correlated input features tended to cause the model to produce weights very close to zero for every cell state, which led to a flat prediction (at zero). The elastic net seemed to solve this problem by introducing a penalty term and shrinking a number of features towards zero while maintaining other weights above zero.

---

## Author Comment (AC3)

**Response to Reviewers for Manuscript: Hydrological Concept Formation inside Long Short-Term Memory (LSTM) networks**

Comments/Text of Referee posted in **black**, our text in **purple**.

**Reviewer 2: Anonymous**

Lees et al. adapted a novel method used in Natural Language Processing, "probe", to examine the internal function of the Long Short Term Memory (LSTM) model in rainfall-runoff predictions. Their results over 669 catchments in Great Britain show a good correlation between the LSTM internal states with re-analysis and independent soil moisture and snow cover products.

I agree with the authors that this paper could be a stepping stone to a myriad of interesting explorations in the field of hydrology. I also appreciate the author's effort in providing additional analysis in the appendices. However, I have some minor comments about some parts of the manuscript, mostly about the clarity and the tone toward traditional hydrologic models.

We thank the reviewer for their summary of the paper, and are glad that our message has been correctly interpreted! By addressing the reviewers suggestions, we have revised our manuscript, and we believe our manuscript has significantly improved

I feel the structure of the Introduction is a bit difficult and redundant for me to follow. I could not get the logical flow here. I found the main objective was stated in both the beginning and the end of the introduction. Why do we need a separate and long paragraph about interpreting machine learning from other fields? This paragraph disrupts my focus on LSTM interpretability.

We apologise for any confusion caused. We clarify our intentions below:

1) The objective outlined at the start of the introduction describes WHAT we hope to achieve. It gives the direction of the paper, describing the research as a hypothesis and a set of questions to answer. The objective outlined at the end of the introduction describes HOW we set about addressing that hypothesis and those questions.

2) The reason for the paragraph outlining the interpretation of ML in other fields is that we are aware that explainable AI and interpretable machine learning is a growing field with much to offer us in Hydrology. We ourselves are using a tool developed elsewhere and applying it to LSTMs trained for rainfall runoff modelling. We think it is important to honestly state that the presented methodology in this manuscript is not our invention and that it is a known technique in other fields of science. Acknowledging this fact, we think it

is important to give context about the state of ML interpretability outside of the field of hydrology.

I think the authors don't have to state that LSTM is the best rainfall-runoff model multiple times in the paper (Introduction and Conclusion). While this statement is still debatable, in my opinion, each rainfall-runoff model has its place in the modeling world. LSTM is increasing its popularity because of its robustness, computational efficiency and accuracy. Period. There is no need for bashing one over another.

We never intended to "bash" one model structure over the other, and have updated several sentences to avoid unfair criticism. We do share a pluralist view on the modelling landscape, where different models should be used for different tasks if they excel at them. Thus, we have tried to make our language as accurate and scientific as possible. To us, this also implies that the strong predictive performance of LSTM-based approaches for discharge modelling should not be played down, while also recognizing that other modelling approaches (such as physically based models) often provide very useful insights into the system being modelled and that system's dynamics. The purpose of this study was to bridge this gap and try to develop a methodology for providing a useful understanding of the LSTM-based approaches.

We believe the following four sentences, might be the cause of the reviewers concern:

1. Introduction: *"LSTMs have demonstrated state-of-the-art performance for rainfall-runoff modelling for a variety of locations and tasks [Frame et al., 2021a; Kratzert et al., 2018, 2019e; Lees et al., 2021b; Ma et al., 2020]"*
2. Conclusion: *"LSTM-based rainfall-runoff models offer good hydrological performance"*

We believe that these are important sentences because they motivate the manuscript, since there is a need to understand what these models are doing *because* of the fact that they offer highly accurate simulations.

3. L268: *"Firstly, the LSTM produces more accurate discharge simulations than any other hydrological model, and so we might expect that the intermediate variables are also better represented by the LSTM.,"*

We believe that this sentence is important because it outlines the logical steps in our thinking:
1) The LSTM is accurate at discharge simulation given meteorological inputs
2) Therefore, we expect the LSTM to have accurate intermediate representations of hydrological stores

In order to address the reviewer's concern, we will update the sentence to read:
*"Firstly, previous studies have demonstrated that the LSTM produces more accurate discharge simulations when benchmarked against other hydrological models, and so we might expect that the intermediate variables and processes are also better represented by the LSTM."*

L282: "*Since the LSTM is often the best performing rainfall-runoff model for discharge (Kratzert et al., 2018, 2019c; Gauch et al., 2021; Frame et al., 2021; Gauch et al., 2020) it makes sense to explore the soil moisture that the LSTM associates with a simulated level of discharge.*"
This sentence adds another logical step to our argument outlined in the previous point:

    3) Therefore, we can extract accurate soil moisture simulations that might be useful for other end users

In order to address the reviewer's concern, we will update this to the following:
"*Since the LSTM is often the most accurate rainfall-runoff model for discharge (Kratzert et al., 2018, 2019c; Gauch et al., 2021; Frame et al., 2021; Gauch et al., 2020) it makes sense to explore the soil moisture that the LSTM associates with a simulated level of discharge.*"

Section 2.3 ERA5-Land Data: there is an imbalance between the descriptions of soil moisture and snow depth. I would expect to see more information about snow depth and its accuracy over GB.
We will update this section with a more complete discussion of the snow depth variable and its usefulness over GB, which now includes the following sentences:

"ERA5-Land data has demonstrated improved representation of snow depth, in part due to the increased spatial resolution which adds value in complex mountainous terrain due to improved representation of the orography, and therefore, a better representation of the surface air temperature (Munoz 2021). To our knowledge there are no direct verifications of the ERA5-Land snow product over the area in Northern Scotland where our snow-depth experiments are conducted. However, there is evidence to suggest that in mid-altitude ranges, the improved spatial resolution of ERA5-Land is a dominant factor in improving snow depth estimates, at least over the ERA5 product (Munoz 2021)."

Figure 2: no y label
We will update this as proposed. (Thanks!)

Figure 5: no y label
We will update this as proposed.

Line 249: I thought there are only two meteorological drivers (temperature and precipitation (line 6))?
Apologies, this is the model version with potential evapotranspiration, it's the same model setup as the 2021 benchmarking study (Lees et al 2021). We will update the associated lines describing model inputs.

Line 268: See the second opinion
"*Firstly, the LSTM produces more accurate discharge simulations than **any other** hydrological model, and so we might expect that the intermediate variables are also better represented by the LSTM.,*"
We believe that this sentence is important because it outlines the logical steps in our thinking:

    4) The LSTM is accurate at discharge simulation given meteorological inputs

5)  Therefore, we expect the LSTM to have accurate intermediate representations of hydrological stores

We will update the sentence to read:
"*Firstly, previous studies have demonstrated that the LSTM produces more accurate discharge simulations when benchmarked against other hydrological models, and so we might expect that the intermediate variables are also better represented by the LSTM.*"

Line 282: See the second opinion
"*Since the LSTM is often the best performing rainfall-runoff model for discharge (Kratzert et al., 2018, 2019c; Gauch et al., 2021; Frame et al., 2021; Gauch et al., 2020) it makes sense to explore the soil moisture that the LSTM associates with a simulated level of discharge.* "

This sentence adds another logical step to our argument outlined in the previous point:

6)  Therefore, we can extract accurate soil moisture simulations that might be useful for other end users

We will update this to the following:
"*Since the LSTM is often the most accurate rainfall-runoff model for discharge (Kratzert et al., 2018, 2019c; Gauch et al., 2021; Frame et al., 2021; Gauch et al., 2020) it makes sense to explore the soil moisture that the LSTM associates with a simulated level of discharge.* "

Line 319: I found a recent paper (Tran et al, Development of a Deep Learning Emulator for a Distributed Groundwater–Surface Water Model: ParFlow-ML. Water. 2021) in which the spatial information is included in the LSTM architecture. Do the authors think the probing technique could be used in this architecture? Can the probing technique map between predicted and observed spatially-distributed soil moisture?

This is a very interesting point, thank you for sharing this paper with us.

The analysis presented in this manuscript has the potential to apply to any matrix that can be mapped to a vector. The approach would be directly applicable if the size of the cell state is the same as the target variable (i.e. if there is a 20x20 pixel grid of soil moisture estimates, the cell state size would need to be a 20x20x{hidden_size} pixel grid of cell-state estimates). Then you can imagine mapping that 20x20x{hs} tensor to the 20x20 pixel grid of soil moisture data to capture the amount of information stored in all of the cell state. We are therefore using the regression to collapse the final dimension (hs) into the 20x20 grid of soil moisture estimates. This is a specific example describing how the method would work so long as you can imagine using a linear regression with the input features mapping to a target variable of interest.

In future work we intend to explore spatially explicit LSTM architectures, like the one you have shown here, and extending our method to these approaches is certainly an interesting area for future work.

---

## Author Response (AR1)

**Editor Comments: Alexander Gruber**

Comments/Text of Referee posted in **black**, our text in **purple**.

We thank the editor for confirming the important aspects of the reviewers comments. We have addressed the following comments and believe that the manuscript is definitely improved!

Dear authors,

thank you for your submission. The referees are, overall, very positive about your manuscript and I share their opinion that the presented study provides an interesting and valuable novel approach to tackle the problem of interpretability in ML for Earth science problems.

I thus invite you to carefully revise your manuscript by implementing your proposed changes to the manuscript, which I believe do, in general, address most of their concerns. I, personally, share the following concerns of the reviewers in particular:

(i) Referee #1: I agree that it might be difficult for readers not too familiar with LSTMs and some other concepts (such as elastic nets) to understand all that is presented. While I also acknowledge that some background is provided and that it is difficult to strike a good balance, I tend to **think that a little more detail (perhaps as am illustrative figure) might be beneficial for a broader readership.**

We have added an Appendix Section F and Figure F1 which give more detail on the LSTM, as well as providing a list of more complete references for readers who may want to further explore the LSTM structure.

(ii) Referee #2: I share the opinion that statements such as "the LSTM produces more accurate discharge simulations than any other hydrological model" need to be toned down a bit. In my view, there isn't a large enough body of evidence that would proof that LSTMs universally perform better than hydrological models, which I don't believe is what the authors want to say, but the text somewhat reads as so…

We agree and have changed our statements as outlined in the responses to Reviewer 2.

L278 "*Firstly, previous studies have demonstrated that the LSTM produces more accurate discharge simulations when benchmarked against other hydrological models, and so we might expect that the intermediate variables are also better represented by the LSTM.*"

L292 "*Since the LSTM was found to be the most accurate rainfall-runoff model for discharge in a series of studies (Kratzert et al., 2018, 2019c; Gauch et al., 2021; Frame et al., 2021; Gauch et al., 2020) it makes sense to explore the soil moisture that the LSTM associates with a simulated level of discharge.*"

Would it perhaps be more fair to say that physically based models should - in theory - be more generalisable (by definition), yet they often require model calibration to account for representativeness errors; whereas ML-based approaches learn whatever data you throw at them and MAY be generalisable provided that actual physical processes are learned? I believe this study makes a good case that it could, and I believe this is also its intention... The fact that a well-trained non-linear ML model might, in the presence of good training data, often - if not always - outperform a perhaps not perfectly calibrated physical model with various simplifying assumptions etc. solely in terms of predictive accuracy is, from how I see it, not really a relevant statement in any context.

Thank you for the effective summary of one of our conclusions. In order to more concretely address this point we have added the following statement in the conclusion:

"*This finding offers a potential explanation for recent research that showed LSTMs have the potential to generalise to out of sample conditions \citep{frame2021deep}, since they have learned physically realistic mappings from inputs to outputs.*"

Best regards,
Alexander Gruber